# Matching patients to clinical trials with large language models

Qiao Jin [1], Zifeng Wang[2], Charalampos S. Floudas [3], Fangyuan Chen [4], Changlin Gong [5], Dara Bracken-Clarke[3], Elisabetta Xue[3], Yifan Yang [1,6], Jimeng Sun [2] & Zhiyong Lu [1] ✉

Patient recruitment is challenging for clinical trials. We introduce TrialGPT, an end-to-end framework for zero-shot patient-to-trial matching with large language models. TrialGPT comprises three modules: it first performs large-scale filtering to retrieve candidate trials (TrialGPT-Retrieval); then predicts criterion-level patient eligibility (TrialGPT-Matching); and finally generates trial-level scores (TrialGPT-Ranking). We evaluate TrialGPT on three cohorts of 183 synthetic patients with over 75,000 trial annotations. TrialGPT-Retrieval can recall over 90% of relevant trials using less than 6% of the initial collection. Manual evaluations on 1015 patient-criterion pairs show that TrialGPT-Matching achieves an accuracy of 87.3% with faithful explanations, close to the expert performance. The TrialGPT-Ranking scores are highly correlated with human judgments and outperform the best-competing models by 43.8% in ranking and excluding trials. Furthermore, our user study reveals that TrialGPT can reduce the screening time by 42.6% in patient recruitment. Overall, these results have demonstrated promising opportunities for patient-to-trial matching with TrialGPT.

Clinical trials examine the effectiveness of medical interventions and provide crucial evidence that can be used to guide clinical practice. They also offer an opportunity for participants to receive experimental treatments that could potentially improve their health outcomes. However, matching patients to suitable clinical trials can be a challenging process[1–3]. This process includes analyzing a patient's medical history, understanding the eligibility criteria of each clinical trial, and ensuring a match that satisfies both patient needs and trial requirements. As such, manually matching patients and clinical trials is often labor-intensive, time-consuming, and prone to human errors.

Recently, artificial intelligence (AI) has shown promise in improving the efficiency and accuracy of patient-trial matching[4,5]. Based on the directionality, there are two types of patient-trial matching tasks. The "trial-to-patient" scheme matches one trial to a list of candidate patients[6], which is a common need for clinical trial organizers and can

be done by converting the trial criteria to structured query languages and searching the patient database[6–8]. On the other hand, the "patient-to-trial" scheme matches one patient to a list of candidate clinical trials[9–13]. In this study, we focus on the patient-centric "patient-to-trial" scheme because such a model can empower individual patients as well as referral offices to explore a large set of potentially eligible clinical trials. However, the heterogeneity and ambiguity inherent in patient records and clinical trial criteria induce significant challenges for AI algorithms. Prior efforts encoded patient records and trial criteria into dense embeddings using neural networks, aiming to represent them in the same embedding space that enables patient-trial matching through similarity search[14–16]. Nonetheless, training neural networks with the language understanding capability of criteria texts and patient records requires large datasets. This is often infeasible due to the lack of paired patient-criterion matching annotations. Moreover, the dense retrieval

[1]National Center for Biotechnology Information (NCBI), National Library of Medicine (NLM), National Institutes of Health (NIH), Bethesda, USA. [2]Department of Computer Science, University of Illinois Urbana-Champaign, Urbana, IL, USA. [3]Center for Immuno-Oncology, Center for Cancer Research, National Cancer Institute, National Institutes of Health, Bethesda, USA. [4]School of Medicine, University of Pittsburgh, Pittsburgh, USA. [5]Jacob Medical Center, Albert Einstein College of Medicine, Bronx, USA. [6]School of Computer Science, University of Maryland College Park, Maryland, USA. ✉e-mail: zhiyong.lu@nih.gov

process is non-explainable, making it challenging to debug and often resulting in skepticism among medical experts when applied to new criteria and patient groups.

In this work, we aim to explore how recent large language models (LLMs) such as GPT-4[17] can aid the process of patient-to-trial matching in a data-efficient and transparent way. LLMs are transformer-based models[18] that can understand a given context and generate human-like responses accordingly. They have shown state-of-the-art capabilities in both the general domain[17,19] and biomedicine[20], including question answering[21-26] and clinical trial design[27,28]. Several pilot studies have also explored using LLMs to enhance the first-stage retrieval of clinical trials by information extraction[29,30], perform data augmentation with synthetic data[31-33], and structuralize clinical trial criteria[34]. In contrast, our study presents an end-to-end solution, i.e., trial retrieval, matching, and ranking, to streamline clinical trial patient recruitment with LLMs.

The proposed method, namely TrialGPT, consists of three key components: TrialGPT-Retrieval, TrialGPT-Matching, and TrialGPT-Ranking. Given a patient note, TrialGPT-Retrieval first locates hundreds of highly relevant candidate clinical trials from a large initial collection using keyword generation and hybrid-fusion retrieval. Based on the retrieved trials, TrialGPT-Matching predicts the criterion-level eligibility of each patient with three elements: natural language explanations showing the relevance of the patient to the criterion, locations of relevant sentences in the patient note that are relevant to the target criterion, and the eligibility classification indicating whether the patient meets this criterion. Finally, TrialGPT-Ranking aggregates the TrialGPT-Matching results at the trial level and uses such scores to get a ranked list of clinical trials based on the eligibility of a given patient.

Our evaluations are conducted with three publicly available cohorts of 183 synthetic patients with over 75,000 trial eligibility annotations. Experimental results show that TrialGPT-Retrieval can recall over 90% of relevant clinical trials using less than 6% of the candidate clinical trials. The model also generates better keywords for trial retrieval compared to those produced by four human clinicians. Three domain experts evaluate TrialGPT-Matching on 1015 patient-criterion pairs, and the results show that TrialGPT-Matching can accurately explain patient-criterion relevance, locate relevant sentences, and predict criterion-level eligibility with an accuracy close to that of human experts. We then evaluate the trial-level scores by TrialGPT-Ranking and results show that they are highly correlated with expert eligibility annotations. Such scores can be used to match eligible trials with patients effectively and exclude ineligible trials, with a performance of 43.8% higher than the best baselines.

We also conducted a pilot user study that mimics the actual clinical trial matching task at the National Cancer Institute (NCI). In the evaluation, each patient-trial pair is evaluated by one medical expert with TrialGPT and another one without TrialGPT. We also ensure that each medical expert annotates half of the pairs with TrialGPT and half without to mitigate the skill differences between the annotators when computing the time reduction. The overall time saving for all patient-trial pairs is about 42.6%, which shows its great potential to enhance the efficiency of the clinical trial matching process.

## Results
### TrialGPT architecture
The architecture of TrialGPT is shown in Fig. 1. Overall, TrialGPT consists of three components: Retrieval (Fig. 1a), Matching (Fig. 1b), and Ranking (Fig. 1c). TrialGPT-Retrieval is designed to filter out most of the irrelevant clinical trials in a large initial collection. Specifically, TrialGPT-Retrieval can generate a list of keywords based on the patient summary and feed them into a hybrid-fusion retriever to obtain a relatively small subset from a potentially large initial collection, while maintaining high recalls of relevant clinical trials. This retrieval step is designed to ensure the scalability of the TrialGPT in real-world applications where there are tens of thousands of clinical trials to consider,

such as matching against as many as 23,000 active clinical trials being conducted in a single country (i.e., the United States).

For each candidate clinical trial returned by the TrialGPT-Retrieval, TrialGPT-Matching analyzes the eligibility of the given patient in a criterion-by-criterion style. For each criterion, TrialGPT-Matching generates three elements: (1) the explanation of the patient-criterion relevance; (2) the locations of relevant sentences in the patient notes to the criterion; (3) the eligibility classification for the patient-criterion pair.

Finally, TrialGPT-Ranking aggregates the criterion-level predictions by TrialGPT-Matching to derive trial-level scores that can be subsequently used to rank clinical trials by eligibility and exclude the ones that are explicitly ineligible.

### Patient cohorts
To evaluate TrialGPT, we use the patient summaries and clinical trials from three publicly available cohorts: a test collection for patient-trial matching published by the Special Interest Group on Information Retrieval (SIGIR) in 2016[10], and the 2021 and 2022 Clinical Trials (CT) tracks[9,13] of the Text REtrieval Conference (TREC). The SIGIR cohort has three different patient-trial eligibility labels: irrelevant ("would not refer this patient for this clinical trial"), potential ("would consider referring this patient to this clinical trial upon further investigation"), and eligible ("highly likely to refer this patient for this clinical trial"). The TREC cohorts also have three different patient-trial eligibility labels: irrelevant ("the patient is not relevant for the trial in any way"), excluded/ineligible ("the patient has the condition that the trial is targeting, but the exclusion criteria make the patient ineligible"), and eligible ("the patient is eligible to enroll in the trial"). We used the combination of the judged clinical trials for all patients in the individual cohort as the considered initial trials for TrialGPT-Retrieval for two reasons: (1) the largest realistic search space for clinical trials is about 23 thousand (the maximum number of active trials in a single country), which is similar to the number of judged trials in the TREC cohorts; (2) these cohorts used pooled judgment, which means that there might be eligible clinical trials outside of the labeled corpus but will be considered as irrelevant if retrieved. The baseline statistics of patient cohorts are shown in Table 1.

### TrialGPT-Retrieval can generate keywords for effective clinical trial filtering
As shown in Fig. 2a, in the retrieval stage, large language models are prompted to generate a list of keywords for the initial screening of clinical trials at scale. For each keyword, we then get the list of relevant clinical trials using a hybrid retriever that matches both lexical and semantic information. The retrieval results are then combined into a ranked list with reciprocal rank fusion. We evaluated the retrieval performance of keywords generated by GPT-4 and GPT-3.5 using all three cohorts. As a comparison, we also tested when the raw patient notes were used to retrieve clinical trials. On the SIGIR cohort, we show the performance of clinician-generated keywords that are annotated in the original dataset.

The recalls of relevant clinical trials at different depths are shown in Fig. 2b. In all three cohorts, keywords generated by GPT-4 and GPT-3.5 consistently achieved the highest performance, while directly using the raw patient notes achieved the lowest performance. This indicates that large language models can effectively generate keywords from patient notes for clinical trial retrieval. In terms of the retriever, the semantic MedCPT is much better than the lexical BM25 retriever, and the hybrid retriever achieves the best performance. In the SIGIR cohort, the performance of clinician-generated keywords is between the LLM-generated keywords and the raw notes. This shows that large language models can already generate better keywords than human clinicians for clinical trial retrieval. Overall, the average recall at the top 500 retrieved clinical trials is 50.0%, 83.4%, 86.2% for the raw note,

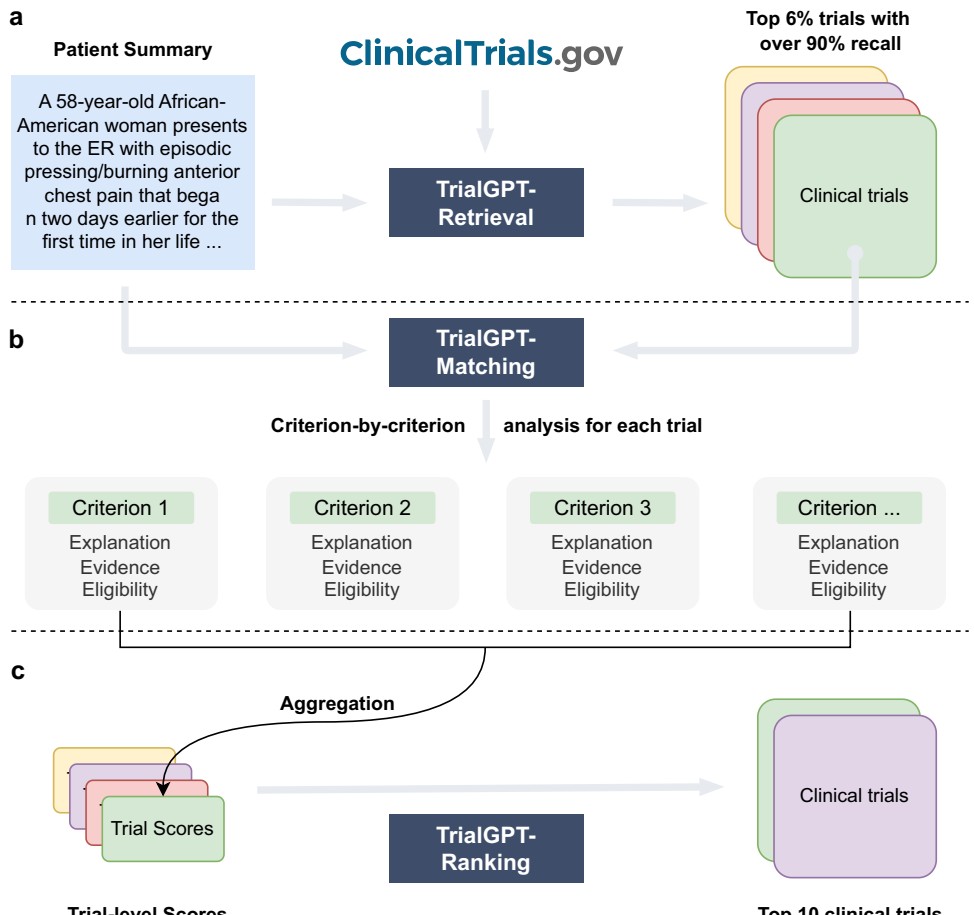

**Fig. 1 | The overall architecture of TrialGPT. a** TrialGPT-Retrieval can filter out most of the irrelevant trials from the initial collection and return a list of candidate clinical trials; **b** For a given patient, TrialGPT-Matching can explain the relevance, generate the evident sentence locations, and predict the eligibility classification for each criterion in a trial; **c** TrialGPT-Ranking can aggregate the criterion-level predictions by TrialGPT-Matching and use these scores to perform fine-grained ranking to get the final recommended trials.

TrialGPT-Retrieval with GPT-3.5, and TrialGPT-Retrieval with GPT-4, respectively. On average, to recall at least 90% of the relevant clinical trials, GPT-4-based TrialGPT-Retrieval only needs to select 5.5% of the initial document collection, and GPT-3.5-based TrialGPT-Retrieval needs 7.0%. As such, TrialGPT-Retrieval significantly improves the

scalability by filtering out most of the irrelevant clinical trials and returning a short list of candidate clinical trials for further finer-grained analyses by TrialGPT-Matching and TrialGPT-Ranking.

## TrialGPT-Matching achieves a high criterion-level prediction accuracy

As shown in Fig. 1b, TrialGPT-Matching first generates the rationales and the relevant sentences for each criterion. Then, it predicts the criterion-level eligibility classification based on the rationales. TrialGPT assigns each inclusion criterion a label within {Included, Not included, Not enough information, Not applicable} and each exclusion criterion a label within {Excluded, Not excluded, Not enough information, Not applicable}. We sampled 105 patient-trial pairs from 53 patients in the SIGIR cohort, which contains 1015 patient-criterion pairs. Three physicians were recruited and manually annotated these pairs regarding the criterion-level output elements by GPT-4: (1) the correctness of TrialGPT relevance explanation between the given patient and the criterion, (2) the relevant sentence locations in the patient note, and (3) the criterion-level prediction of the given patient's eligibility. Consensus annotations derived from individual annotations and further discussions are used as the ground truth.

Evaluating relevance explanations: We show the percentage of "correct", "partially correct" and "incorrect" TrialGPT explanations in Fig. 3a. Overall, most explanations are "correct" (87.8%) by manual evaluations, while less than 10% of explanations are "partially correct" (9.66%), and only a small proportion are "incorrect" (2.56%). We also

**Table 1 | Baseline statistics of the three patient cohorts used in this work**

| Cohort | SIGIR | TREC 2021 CT | TREC 2022 CT |
|---|---|---|---|
| N | 58 | 75 | 50 |
| Age (year) | 38.5 ± 23.7 | 41.6 ± 19.4 | 35.3 ± 20.2 |
| Sex (male: female) | 29: 29 | 38: 37 | 28: 22 |
| Note length (words) | 88.7 ± 36.8 | 156.2 ± 45.4 | 109.9 ± 21.6 |
| Eligible trials/patient | 7.3 ± 6.7 | 74.3 ± 49.0 | 78.8 ± 67.3 |
| Potential trials/ patient | 11.7 ± 10.2 | None | None |
| Excluded trials/ patient | None | 80.3 ± 60.3 | 60.7 ± 65.5 |
| Irrelevant trials/ patient | 47.1 ± 19.5 | 323.2 ± 93.2 | 568.4 ± 164.1 |
| Considered initial trials | 3621 | 26149 | 26581 |

We show the mean ± standard deviation for applicable variables. "None" denotes there is no such eligibility label in the corresponding cohort. SIGIR: the patient-trial matching cohort published at the Special Interest Group on Information Retrieval (SIGIR).
*TREC* the Text REtreival Conference (TREC), *CT* the clinical trials track at TREC.

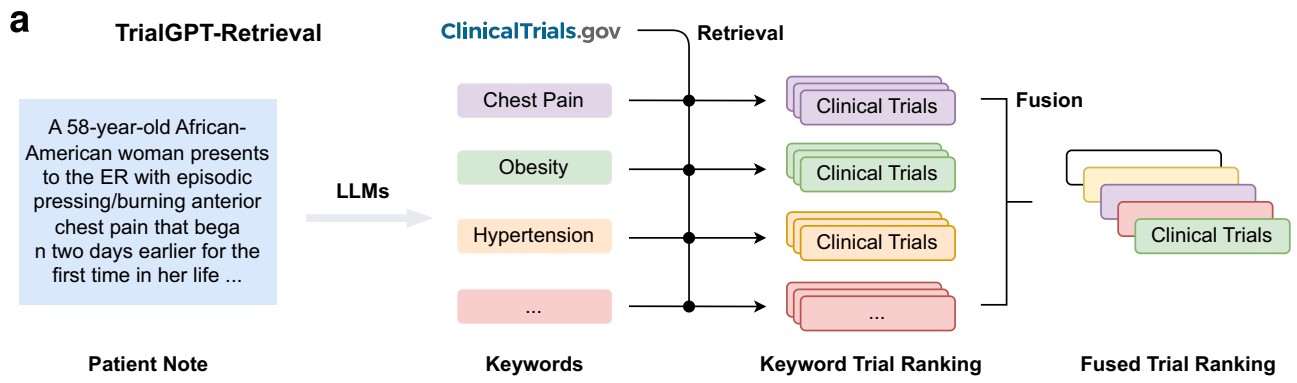

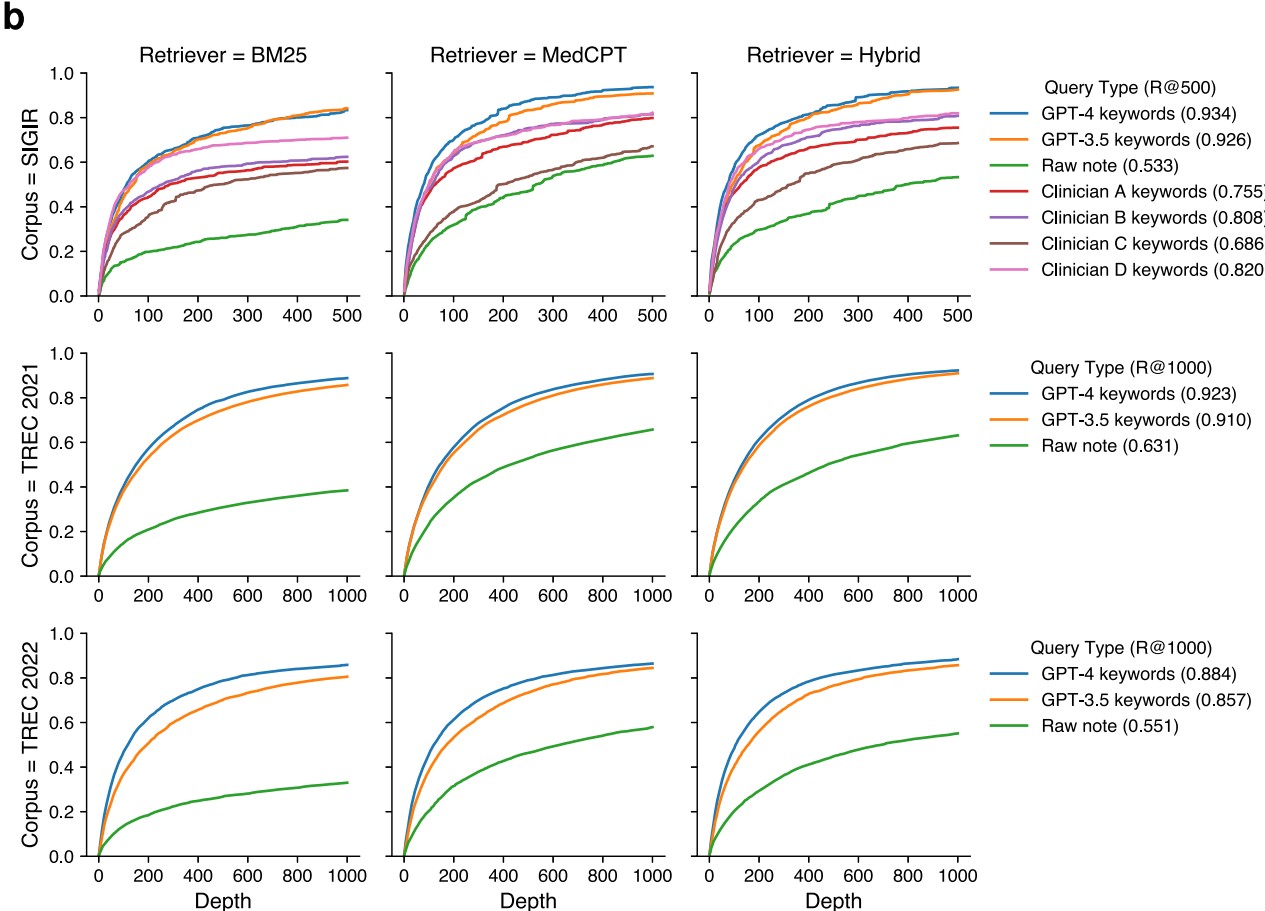

**Fig. 2 | First-stage retrieval results. a** Overview of TrialGPT-Retrieval. LLMs first generate a list of keywords for a given patient note. These keywords are used to derive the keyword-level relevant clinical trials, which are then fused to generate a final ranking. **b** Recalls of relevant clinical trials at different depths for various query types and retrievers. The hybrid retriever combines the results of the BM25 (lexical matching) and the MedCPT (semantic matching) retrievers. Source data are provided as a Source Data file.

found that most of the incorrect explanations are for criteria labeled as "not included" and "not excluded", which usually require implicit inference. TrialGPT exhibits much fewer mistakes when the criteria are explicitly "included" or "excluded". These results suggest that TrialGPT can effectively explain how a patient is relevant to an eligibility criterion.

Evaluating relevant sentence locations: We compare the relevant sentences predicted by TrialGPT-Matching against the ground-truth expert annotations. As shown in Fig. 3b, the TrialGPT-predicted sentence locations are 90.1% correct (precision) and cover 87.9% of the ground-truth relevant sentence IDs (recall), leading to an F1 score of

88.6%. The performance of TrialGPT is close to that of human experts, ranging from 86.9% to 91.5%. This shows that TrialGPT can faithfully locate relevant sentences in patient notes, providing strong explainability for human expert use and oversight.

Evaluating eligibility prediction: Finally, we evaluate the criterion-level eligibility labels predicted by TrialGPT and expert annotators against the ground-truth annotations. Figure 3c and d show the confusion matrices for these predictions. Overall, TrialGPT-Matching achieves a prediction accuracy of 0.873, close to the expert performance (0.887–0.900). For the inclusion criteria, TrialGPT reaches a prediction accuracy of 0.899 for all four labels, which is within the

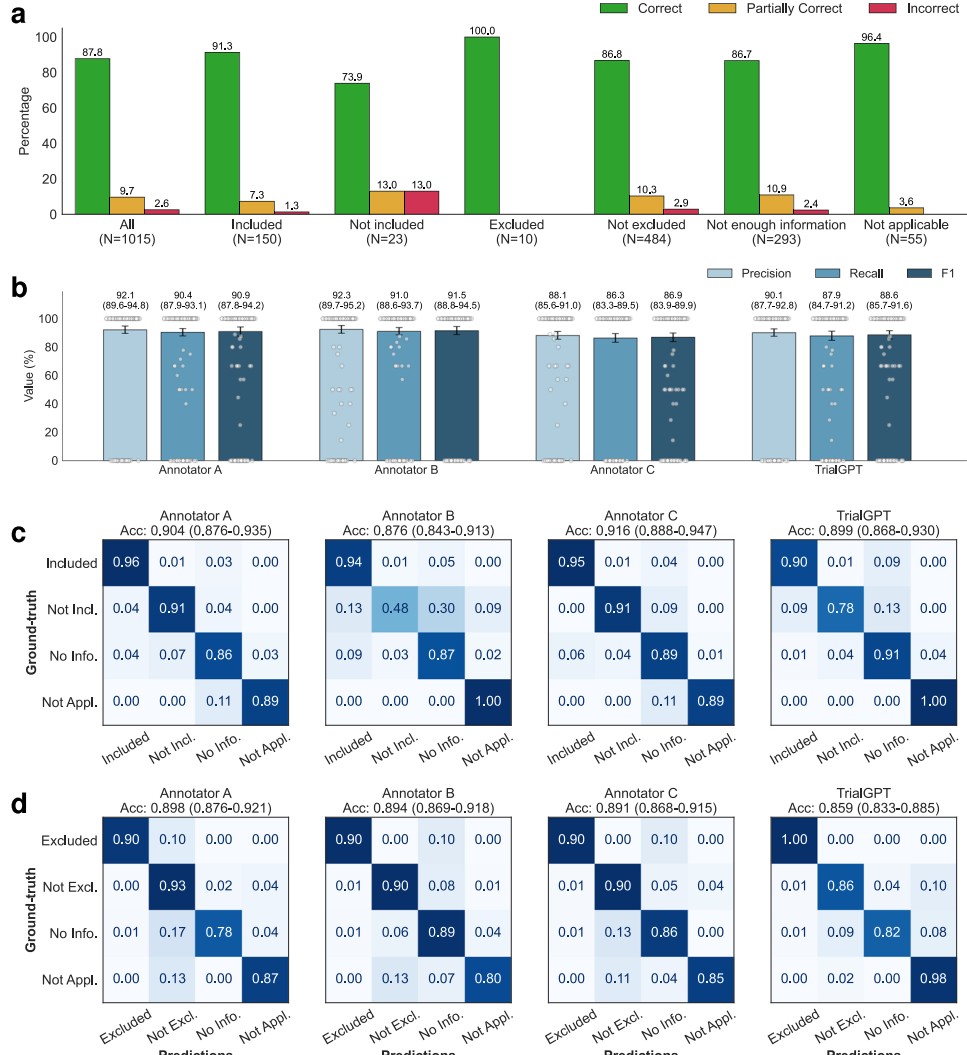

**Fig. 3 | Manual evaluations of criterion-level predictions by GPT-4-based TrialGPT-Matching. a** The percentage of correct, partially correct, and incorrect relevance explanations generated by TrialGPT-Matching; **b** Evaluation results of the relevant sentences located by TrialGPT-Matching based on 405 criterion-level annotations where at least one sentence is labeled as relevant in the ground-truth. 95% confidence intervals estimated by bootstrapping are shown as error bars; **c** The confusion matrices of the eligibility for inclusion criteria predicted by human experts and TrialGPT-Matching; **d** The confusion matrices of the eligibility for exclusion criteria predicted by human experts and TrialGPT-Matching. Not Incl.: Not included. Not Excl.: Not excluded. No Info.: Not enough information. Not Appl.: Not applicable. Source data are provided as a Source Data file.

experts' accuracy range from 0.876 to 0.916. For the exclusion criteria, while the accuracy is high for criteria labeled with "excluded" (1.00) and "not applicable" (0.98), TrialGPT tends to confuse among "not excluded", "no relevant information", and "not applicable". TrialGPT achieves an accuracy of 0.859 on the exclusion criteria. These results suggest that TrialGPT can accurately predict patient eligibility at the criterion level, with a performance close to that of human experts.

We further inspected the 26 criterion-level predictions that are labeled as "Incorrect" by annotator consensus. Four types of errors have been identified: (E1) Incorrect reasoning, where TrialGPT predicts "not enough information" but the matching result can be implicitly inferred; (E2) Lack of medical knowledge, such as not recognizing that one medical term is synonymous with another or that one term is a subtype of another; (E3) Ambiguous label definitions, confusing "not enough information" with "not applicable" due to unclear or overlapping label definitions; (E4) Other unclassified errors. Supplementary Table 1 shows the proportion and example of each error type made by TrialGPT. Most (30.7%) of the errors are due to incorrect reasoning, followed by ambiguous or redundant definitions of the eligibility classes (26.9%). Lack of medical knowledge contributes to

about 15.4% of the total errors. These results suggest that improving the medical capabilities of the backbone LLM is an important future direction.

## TrialGPT-Ranking scores correlate with trial-level eligibility

TrialGPT-Matching has achieved high prediction accuracy at the criterion level. However, since one clinical trial typically has many inclusion and exclusion criteria, trial-level scores should be computed to decide to which extent a given patient is eligible or ineligible. As such, the third component, TrialGPT-Ranking, aggregates the criterion-level predictions into trial-level scores, as shown in Fig. 1c. In this section, we analyze the correlations between patient-trial eligibility and eight variants of trial-level scores, which are computed by two types of methods: linear aggregations and LLM aggregations. For each patient in the three cohorts, we consider the top 500 clinical trials returned by TrialGPT-Retrieval for the analysis. The results are presented as box plots in Fig. 4.

Linear aggregations: Six scores are computed by counting the percentages of six different criterion-level eligibility predictions of TrialGPT (details in Methods). Their correlations with different types of

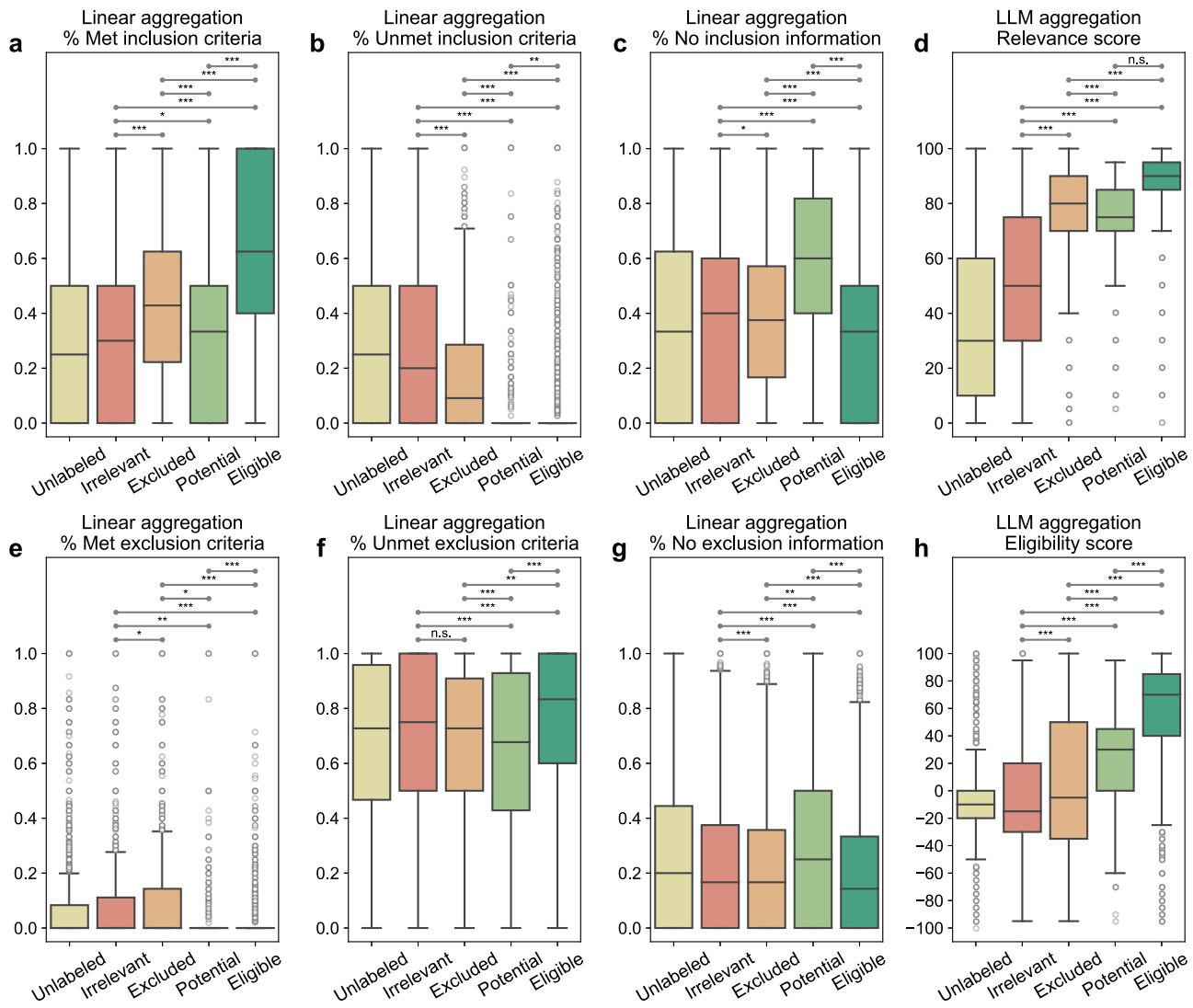

**Fig. 4 | Correlation between differently aggregated TrialGPT scores and the ground-truth patient-trial eligibility labels. a** The percentage of inclusion criteria predicted as "included" by TrialGPT; **b** The percentage of inclusion criteria predicted as "not included"; **c** The percentage of inclusion criteria predicted as "no relevant information"; **d** The LLM-aggregated relevance score; **e** The percentage of exclusion criteria predicted as "excluded"; **f** The percentage of exclusion criteria predicted as "not excluded"; **g** The percentage of exclusion criteria predicted as "no relevant information"; **h** The LLM-aggregated eligibility score. "*" denotes $p < 0.05$, "**" denotes $p < 0.01$, "***" denotes $p < 0.001$, and "n.s." denotes not significant ($p > 0.05$) by two-sided independent t-test. There are 60,240 unlabeled, 15,459 irrelevant, 6981 excluded, 647 potential, and 8173 eligible patient-trial pairs. Center line, median; box limits, upper and lower quartiles; whiskers, 1.5x interquartile range; points, outliers. Source data are provided as a Source Data file, including all the exact p values.

trial-level eligibility labels are shown in Fig. 4a–c for inclusion criteria and Fig. 4e–g for exclusion criteria. Figure 4a shows the percentage of inclusion criteria predicted as "included" by TrialGPT. As expected, Fig. 4a implies that the patients meet the highest percentages of inclusion criteria in eligible clinical trials and meet the lowest percentages of inclusion criteria in irrelevant clinical trials. The percentage of met inclusion criteria falls in between for relevant but ineligible trials. Figure 4b shows the percentage of inclusion criteria predicted as "not included", which approximately follows the reverse trends of the met inclusion criteria (Fig. 4a). Noticeably, no inclusion criterion is classified by TrialGPT-Matching as "not included" in most of the eligible patient-trial pairs, confirming the correctness of the model. Figure 4d shows the percentage of exclusion criteria predicted as "excluded". Importantly, patients meet more exclusion criteria in ineligible clinical trials than in eligible clinical trials. Similar to Fig. 4b, no exclusion criteria are labeled as met by TrialGPT-Matching in most eligible and potentially eligible patient-trial pairs. This is a characteristic feature of

patient-trial pairs that are explicitly excluded and can be exploited in patient-trial matching.

LLM aggregations: We also use LLMs to further aggregate the criterion-level predictions of TrialGPT, resulting in the general relevance and the eligibility scores (details in Methods). The general relevance score (0–100) is shown in Fig. 4d, where the irrelevant patient-trial pairs are much lower than the other labeled groups. Eligible and ineligible/potential patient-trial groups have certain overlaps, but the former is still significantly higher than the latter. The eligibility score (-100–100) is shown in Fig. 4h, where negative scores denote ineligible, positive scores denote eligible, and a score of 0 denotes neutral. Overall, the trends of LLM-aggregated eligibility scores follow those of the LLM-aggregated relevance score. Noticeably, the eligible patient-trial pairs and the excluded (but relevant) patient-trial pairs have the least distributional overlap in TrialGPT-aggregated eligibility score, which indicates its superiority in distinguishing these two groups.

**Table 2 | Performance of different methods for ranking and excluding clinical trials**

| Application | | | Ranking | | Excluding | Overall |
|---|---|---|---|---|---|---|
| Method / Metric | | | NDCG@10 | P@10 | AUROC | Average |
| SciFive[46] (encoder-decoder) | Further trained on MedNLI[37] | | 0.4271 | 0.3787 | 0.5895 | 0.4652 |
| BioBERT[47] (dual-encoder) | Further trained[48] on MNLI[49], SNLI[50], SciNLI[51], SciTail[52], MedNLI[37], and STSB[53] | | 0.4091 | 0.3746 | 0.5952 | 0.4596 |
| PubMedBERT[54] (dual-encoder) | | | 0.4327 | 0.3874 | 0.5976 | 0.4726 |
| SapBERT[55] (dual-encoder) | | | 0.4148 | 0.3738 | 0.5933 | 0.4606 |
| BioLinkBERT[36] (cross-encoder) | Further trained on MedNLI[37] | | 0.4797 | 0.4281 | 0.6176 | 0.5085 |
| TrialGPT-Ranking (GPT-3.5) | Feature combination | | 0.5395 | 0.5115 | 0.6582 | 0.5697 |
| TrialGPT-Ranking (GPT-4) | Linear aggr. | Sign(% Included) | 0.6097 | 0.5653 | 0.6765 | 0.6172 |
| | | Sign(% Not included) | 0.4923 | 0.4556 | 0.6652 | 0.5377 |
| | | Sign(% Excluded) | 0.3930 | 0.3715 | 0.6477 | 0.4707 |
| | | Sign(% Not excluded) | 0.4130 | 0.3886 | 0.5820 | 0.4612 |
| | LLM aggr. | Sign(Relevance) | 0.7281 | 0.6700 | 0.7402 | 0.7128 |
| | | Sign(Eligibility) | 0.7252 | 0.6724 | 0.7895 | 0.7290 |
| | Feature combination | | 0.7275 | 0.6688 | 0.7979 | 0.7314 |

The Sign() function assigns suitable signs for the corresponding task, e.g., for "% Included", it will be "+" for ranking and "-" for excluding clinical trials. Aggr.: aggregation. NDCG@10: normalized discounted cumulative gain at 10. P@10: precision at 10.
*AUROC* the area under the receiver operating characteristic curve.

In summary, criterion-level TrialGPT-Matching predictions can be aggregated into trial-level scores by TrialGPT-Ranking that are highly correlated with patient-trial eligibility. The results of linear aggregations demonstrate that eligible patient-trial pairs have the highest proportions of met inclusion criteria and unmet exclusion criteria, while ineligible patient-trial pairs have the highest proportions of met exclusion criteria. In addition, the LLM aggregations are also significantly correlated with the manual eligibility labels. These results suggest that the aggregated scores of TrialGPT can be used to rank or exclude clinical trials for patient recruitment.

## TrialGPT-Ranking can effectively rank and exclude candidate clinical trials

In this section, we evaluate TrialGPT by ranking candidate clinical trials and excluding ineligible clinical trials for given patients (component Fig. 1c). Based on the correlation analysis, we design a suite of scoring methods to aggregate criterion-level predictions of TrialGPT to generate a trial-level score for ranking the candidate trials. Similarly, we consider the top 500 clinical trials returned by TrialGPT-Retrieval for each patient in the three cohorts Table 2 shows the Normalized Discounted Cumulative Gain at rank 10 (NDCG@10), Precision at rank 10 (P@10), and Area Under the Receiver Operating Characteristic curve (AUROC) of different methods in comparison to state-of-the-art models, which are described in Methods.

Ranking candidate clinical trials: As shown in Table 2, TrialGPT-Ranking outperforms all compared baselines, including dual-encoder, cross-encoder, and encoder-decoder models trained on different biomedical and clinical natural language inference (NLI)[35] datasets. The best baseline for ranking clinical trials is the cross-encoder BioLinkBERT[36] trained on MedNLI[37], which achieves the NDCG@10 of 0.4797 and the P@10 of 0.4281. The most effective features of GPT-4-based TrialGPT for ranking are the LLM-aggregated eligibility scores. They achieve NDCG@10 of 0.7252 and P@10 of 0.6724, which are much higher than other aggregations. Combining both linear and LLM aggregations yields the highest NDCG@10 performance of 0.7275 and the P@10 of 0.6699 for GPT-4-based TrialGPT-Ranking. While GPT-3.5-based TrialGPT-Ranking also surpasses all baselines, the improvements are not as significant as GPT-4-based TrialGPT-Ranking. It should be noted that the results are not directly comparable to the results of TREC CT participating systems as we used the initial corpora of more realistic sizes and reported the average performance on three different cohorts.

Excluding ineligible clinical trials: Table 2 also shows the AUROC of excluding candidate trials, which is modeled as a binary classification task. Similarly, the best baseline for excluding clinical trials is the cross-encoder BioLinkBERT[36] trained on MedNLI[37], achieving the AUROC of 0.6176. This result only shows marginal improvement over the random score baseline, indicating that the task of excluding ineligible trials presents significant challenges. Unlike in the task of ranking clinical trials, the percentage of inclusion criteria predicted as "not included" and the percentage of exclusion criteria predicted as "excluded" also achieve comparable AUROC individually. Again, both GPT-3.5-based and GPT-4-based TrialGPT-Ranking outperforms all baselines, and the combination of the features achieves an AUROC of 0.6582 and 0.7979, respectively.

On average, the feature combination of GPT-4-based TrialGPT-Ranking achieved a performance of 0.7314, which is about 43.8% better than the best-performing baseline score of 0.5085. These experimental results show that TrialGPT can effectively rank candidate clinical trials and exclude ineligible clinical trials, which could facilitate the trial-matching process.

## TrialGPT can reduce the screening time for patient-trial matching

To evaluate whether TrialGPT can assist clinicians in performing the patient recruitment task, we set up a user evaluation that mimics patient-trial matching in real clinical settings. The evaluation result is shown in Fig. 5a. Here, we consider six oncology clinical trials. Among these trials, four are conducted at the National Cancer Institute and have a physician coauthor (C.F.) as principal or associate investigator, and the other two are highly related. The physicians also created six semi-synthetic clinical vignettes (cases 1 to 6, available in the Supplementary Note) based on actual, anonymized patient referral requests. Cases 1–3 are short summaries of patients, while cases 4–6 are more extensive descriptions. For each patient-trial combination, the matching task is to quickly screen the eligibility criteria to either reject the patient or include the patient for further investigation. Two medical doctors (Annotators X and Y) performed the tasks, where half of the patient-trial pairs have TrialGPT predictions, and the other half do not have them. We also ensure that for each patient-trial pair, one annotator screened with TrialGPT, and another without. The accuracy of annotations with TrialGPT is higher than without (97.2% vs. 91.7%), although both are above 90% and their

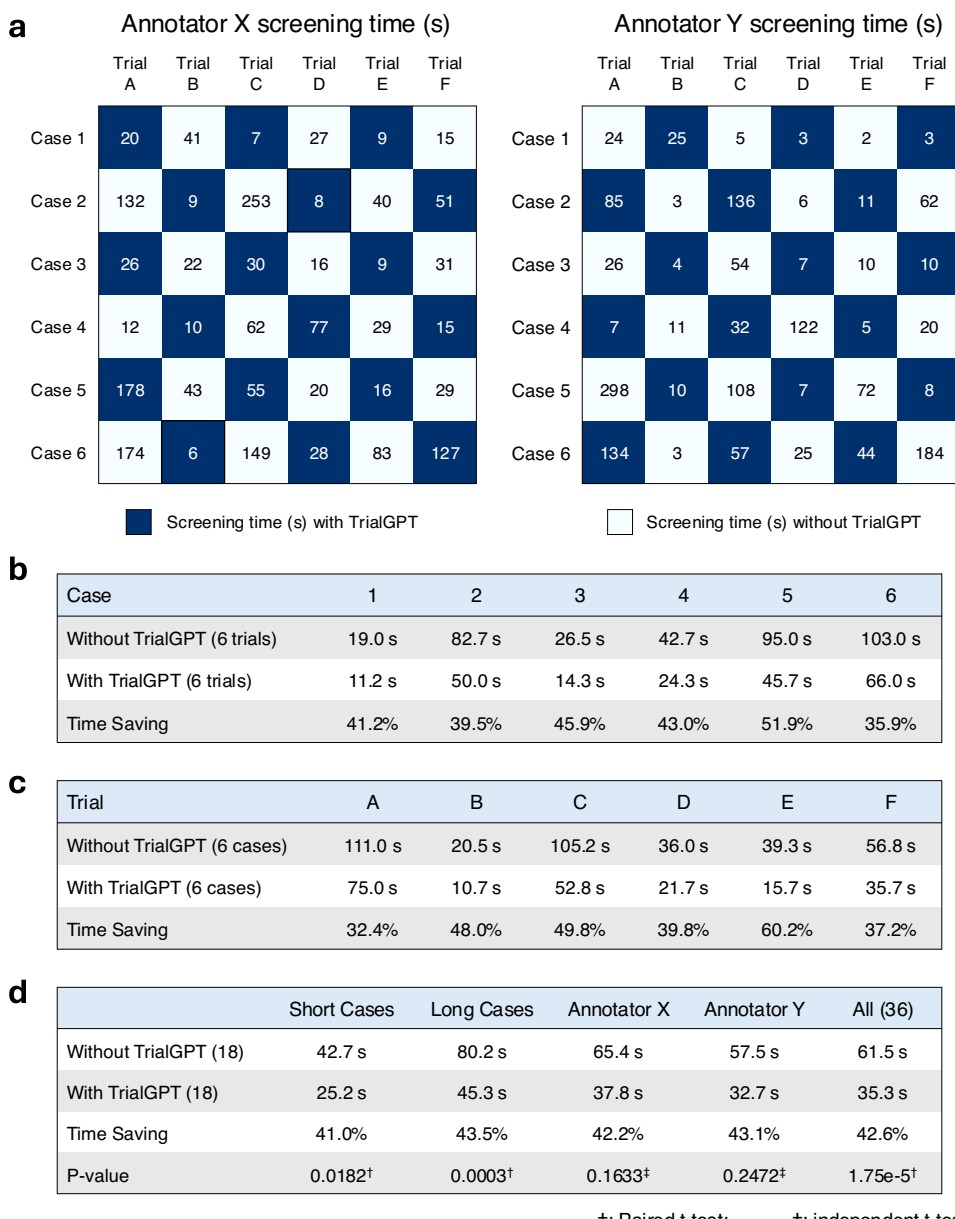

**Fig. 5 | Results of the patient-trial matching user study. a** Experimental design and actual screening times of each patient-trial pair by two expert annotators; **b** Comparison of screening time aggregated by different clinical trials; **c** Comparison of screening time aggregated by different patient cases; **d** Comparison of screening time aggregated by short cases, long cases, annotators, and all pairs. Numbers in parentheses denote the sample sizes in the corresponding group of the comparison. Within the annotator (e.g., Annotator X or Y), significant tests are conducted by two-sided independent t-test. Other significant tests are two-sided paired t-tests. Trial A, B, C, D, E, and F denote NCT04432597, NCT05012098, NCT04287868, NCT04847466, NCT04719988, and NCT04894370, respectively. Source data are provided as a Source Data file.

differences are not statistically significant. The comparison results for matching efficiency are shown in Fig. 5b. There is no significant time difference between Annotator X and Annotator Y in screening with TrialGPT (32.7 s v.s. 37.8 s, $p = 0.73$) or without TrialGPT (57.5 s v.s. 65.4 s, p = 0.75) by two-sided independent t-test, which indicates that they have similar baseline screening capabilities. We observe a consistent trend of improved efficiency with TrialGPT: 35.9% to 51.9% less time is spent for different patients, and 32.4% to 60.2% less time is spent among different trials. In general, more time saving is observed for long cases (43.5%) than for short cases (41.0%). The overall time saving for all patient-trial pairs is about 42.6%, which can greatly improve the efficiency of a team evaluating patient referrals for clinical trials.

## Discussion

We propose TrialGPT, a framework for end-to-end patient-trial matching with LLMs. The technical novelty of TrialGPT is to utilize LLMs for three vital sub-tasks: (1) First-stage retrieval: TrialGPT-Retrieval can filter out most of the irrelevant clinical trials from a large initial collection containing tens of thousands of clinical trials; (2) Criterion-level prediction: TrialGPT-Matching can match patient and clinical trials at criterion-level with explanations, where previous models suffer from the lack of annotated instances and cannot provide explanations; (3) Aggregation of criterion-level predictions: TrialGPT-Ranking further utilizes LLMs to aggregate the criterion-level predictions to generate trial-level scores, which outperforms other linearly aggregated scores at ranking and excluding candidate clinical trials.

The language understanding capability of LLMs is exploited in all tasks, and the language generation capability of LLMs lays the foundation for the explainability of the overall workflow. Compared to the structurization approach adopted by other studies[7,34], TrialGPT utilizes LLMs to analyze the patient summary and the criteria in natural language, which does not require the criteria to follow any formats and is thus more flexible.

For evaluation, we use three publicly available patient-trial matching datasets, where the patients are represented by a paragraph of free-text clinical summary. However, clinical trial matching sometimes requires the recruiters to check the patients' information more comprehensively, which involves longitudinal clinical notes, lab values, and even imaging data. This requires the model to (1) attend to much longer contexts, (2) process structured data, and (3) process multi-modal inputs. These aspects have not been evaluated by this study but are worth exploring in future work. Additionally, future evaluations should investigate how TrialGPT performs when integrating data from electronic health records (EHRs), which often include a combination of structured and unstructured data sources. The ability to seamlessly incorporate such diverse data types would significantly enhance the real-world applicability and increase the validation sample size of our framework. Further, the SIGIR and TREC datasets focus on the semantics of the clinical trial inclusion/exclusion criteria, excluding factors such as geolocation and trial recruitment status as these are addressable through traditional structured query approaches. As a result, any use of TrialGPT would similarly need to ensure the identified trials are appropriate for a patient along these lines as well.

Our work does not justify the position that clinical trial matching should be fully automatic and exclude human recruiters. Experts should always be in the loop of medical AI deployments, and the TrialGPT matching results are only used to assist them in improved efficiency. Evaluation in real-life clinical trial matching scenarios should also focus more on efficiency improvement for human recruiters, instead of solely reporting the prediction performance. In this sense, the explanation capability of TrialGPT, or more generally LLMs, is particularly helpful. This is exemplified by our pilot user study showing that TrialGPT can significantly reduce 42.6% of the screening time on average.

While our experiments have shown promising results in patient-to-trial matching with TrialGPT, this study has three main limitations. First, TrialGPT relies on OpenAI's GPT series LLMs such as GPT-3.5 and GPT-4 as the backbone model. Although GPT-4 is currently the most capable LLM, it is closed-source and can only be accessed via commercial applications or API. Future studies should explore using and fine-tuning other open-source LLMs as alternatives. Second, while our study proposes a novel framework of patient-to-trial matching with LLMs, there are various other prompting strategies for each of the TrialGPT components that are worth exploring in future work. Third, our pilot user study is of limited scope in sample size. Nonetheless, it offers insights into the potential benefits of LLMs for assisting clinical trial matching and provides impetus to conduct larger-scale prospective evaluations regarding the impact of LLM-assisted clinical workflows in future studies.

In summary, we present TrialGPT, a novel architecture that uses large language models to perform patient-trial matching. Our evaluations show that TrialGPT can effectively recall relevant clinical trials in a large-scale collection and accurately predict criterion-level eligibility with faithful explanations. We also explore different pooling methods to aggregate the criterion-level predictions to trial-level scores that can be used to rank or exclude a list of candidate trials. The pilot user study has clearly demonstrated that TrialGPT significantly reduces the screening time needed for human experts. As such, we anticipate that large language models can be valuable in assisting the process of patient-trial matching.

## Methods

### Patient cohorts
We use three publicly available cohorts in this study: the SIGIR 2016 cohort, the TREC 2021 CT cohort, and the TREC 2022 CT cohort. All three cohort annotations only use the patient note and eligibility criteria without considering the geolocations and recruitment status of the clinical trials. Their baseline characteristics are shown in Table 1.

### SIGIR 2016
The original cohort contains 60 patient case reports, but 1 report is removed since the report is about a group of patients (topic ID 201426, "A group of 14 humanitarian service workers…"). Additionally, another case (topic ID 201428) was removed since the there are no relevance judgements for the patient. The patient notes are derived from the Clinical Decision Support (CDS) tracks in TREC 2014[38] and 2015[39], which are "medical case narratives created by expert topic developers that will serve as idealized representations of actual medical records"[38]. They typically describe the patient's medical history, current symptoms, tests, eventual diagnosis, and treatments. Given a patient note, four medical assessors annotate each candidate clinical trial with three possible labels: (a) "would not refer this patient for this clinical trial"; (b) "would consider referring this patient to this clinical trial upon further investigation"; and (c) "highly likely to refer this patient for this clinical trial". We consider label a to be "irrelevant", label b to be "potential", and label c to be "eligible". The candidate clinical trials for annotation are derived from pooling various retrieval methods.

### TREC 2021/2022 CT
The TREC 2021 and 2022 CT tracks contain 75 and 50 patients, respectively. These patient notes are synthetic patient case descriptions, "such as what may be included as part of an admission note". For each patient, they annotate three eligibility labels for the candidate clinical trial: irrelevant ("the patient is not relevant for the trial in any way"), excluded/ineligible ("the patient has the condition that the trial is targeting, but the exclusion criteria make the patient ineligible"), and eligible ("the patient is eligible to enroll in the trial"). The candidate clinical trials are pooled from the submission systems of TREC participants.

### TrialGPT
TrialGPT is an architecture for patient-trial matching with large language models. It is composed of three modules: (1) TrialGPT-Retrieval for filtering out most of the irrelevant clinical trials for a given patient; (2) TrialGPT-Matching for criterion-by-criterion analysis of a given patient-trial pair; and (3) TrialGPT-Ranking for fine-grained re-ranking of a list of candidate clinical trials. TrialGPT is LLM-agnostic, meaning it can be plugged into different backbone LLMs. In this study, we mainly use the GPT-4 (model version: 0613) and GPT-3.5 API (model version: 0613) through Microsoft Azure's OpenAI services. We used the openai package (version 1.30.5) in Python to call the API. In the manuscript, the default backbone LLM of TrialGPT is GPT-4 unless otherwise specified. We set the inference temperature to 0 for deterministic outputs.

### TrialGPT-Retrieval
Given a free-text patient summary, TrialGPT-Retrieval first generates a list of keywords using large language models. These keywords are intended to filter out most of the irrelevant clinical trials in the initial collection. The prompt for keyword generation is shown in Supplementary Table 2. Denote the list of keywords generated for the patient as $[w_1, w_2, \ldots, w_K]$, where $K$ is the number of generated keywords and is set up to 32 in the prompt. The LLMs are also instructed to rank the keywords by importance.

We first conduct hybrid retrieval for each keyword to find relevant clinical trials. Specifically, for keyword $w_i$, we send it to both the traditional BM25 retriever[40] for lexical retrieval and the dense MedCPT

retriever[41] for semantic retrieval. For each considered clinical trial $t_j$, its rank in terms of relevance from BM25 and MedCPT retriever are denoted as $\text{Rank}(\text{BM25}, w_i, t_j)$ and $\text{Rank}(\text{MedCPT}, w_i, t_j)$, respectively. For example, $\text{Rank}(\text{BM25}, w_i, t_j) = 1$ means that clinical trial $t_j$ is the most relevant (ranking the first) for the keyword $w_i$ returned by the BM25 retriever. We use reciprocal rank fusion[42] between the MedCPT and BM25 retrievers for the same keyword, and a decaying weight for combining the scores across different keywords. Finally, the TrialGPT-Retrieval score $s_j$ for a clinical trial $t_j$ for the given patient can be calculated as:

$$s_j = \sum_{\text{Ret}} \sum_{i=1}^{K} \frac{1}{i \times (\text{Rank}(\text{Ret}, w_i, t_j) + C)} \quad (1)$$

where the inner sum is the fusion of different keywords with the decaying factor of $1/i$, the outer sum is the combination of two retrievers ($\text{Ret} \in \{\text{BM25}, \text{MedCPT}\}$), and $C$ is the constant used in reciprocal rank fusion, which we set to 20.

The clinical trials are then ranked by the TrialGPT-Retrieval score and the highest ranked ones are considered as the candidate clinical trials.

## TrialGPT-matching

Here we denote a patient note as a list of $P$ sentences $[s_1, s_2, \ldots, s_P]$, a clinical trial as composed of the background information B (containing the title, conditions, interventions, and the brief summary), a list of $M$ inclusion criteria $[i_1, i_2, \ldots, i_M]$, and a list of $N$ exclusion criteria $[e_1, e_2, \ldots, e_N]$. The objective of this module is to output a free-text relevance explanation $R$, a list of relevant sentence IDs $S$, and the eligibility prediction $E$ for each criterion based on the input patient note. For an inclusion criterion,

$$E \in \{\text{included, not included, not enough information, not applicable}\} \quad (2)$$

while for an exclusion criterion,

$$E \in \{\text{excluded, not excluded, not enough information, not applicable}\} \quad (3)$$

We use different label sets for inclusion and exclusion criteria because the latter are often ambiguous. For example, exclusion criteria of "Pregnancy," "The patient should not be pregnant," and "Pregnant patients will be excluded" serve the same purpose. Traditional entailment labels might not be suitable to distinguish the semantic differences, while our eligibility-oriented label sets provide an end-to-end solution.

We make two LLM inference calls for each patient-trial pair: one for all inclusion criteria, and another one for all exclusion criteria. Overall, the prompt includes the task description, the clinical trial background information B, and the inclusion ($[i_1, i_2, \ldots, i_M]$) and exclusion ($[e_1, e_2, \ldots, e_N]$) criteria. Motivated by chain-of-thought prompting[43], we prompt the model to first generate the relevance explanation as grounding for future predictions of the relevant sentence IDs and the eligibility labels. In addition, we also prompt the model to generate criterion-level predictions in the JSON format, which can be easily parsed for aggregations. The TrialGPT prompts are shown in Supplementary Tables 3 and 4.

## TrialGPT-ranking

After getting the criterion-level predictions from TrialGPT-Matching, TrialGPT-Ranking then aggregates such scores to generate a trial-level score that can be used for practical applications such as ranking and excluding clinical trials. Specifically, we denote the eligibility predictions of TrialGPT for the inclusion criteria and exclusion criteria as

$[E(i_1), E(i_2), \ldots, E(i_M)]$ and $[E(e_1), E(e_2), \ldots, E(e_N)]$, respectively. $M$ is the number of inclusion criteria and $N$ is the number of exclusion criteria in the trial.

Linear aggregation: six scores are simply derived based on the percentages of different eligibility predictions. While more sophisticated scoring methods can be used, we intentionally use these simple and linear aggregation strategies for better probing the capabilities of LLMs.

For a trial's inclusion criteria:

$$\% \text{ met inclusion criteria} = \frac{|\{E(i_x) = \text{included} \mid x = 1, 2, \ldots, M\}|}{M'} \quad (4)$$

$$\% \text{ unmet inclusion criteria} = \frac{|\{E(i_x) = \text{not included} \mid x = 1, 2, \ldots, M\}|}{M'} \quad (5)$$

$$\% \text{ not enough information} = \frac{|\{E(i_x) = \text{not enough information} \mid x = 1, 2, \ldots, M\}|}{M'} \quad (6)$$

$$M' = M - |\{E(i_x) = \text{not applicable} \mid x = 1, 2, \ldots, M\}| \quad (7)$$

For a trial's exclusion criteria:

$$\% \text{ met exclusion criteria} = \frac{|\{E(e_y) = \text{excluded} \mid y = 1, 2, \ldots, N\}|}{N'} \quad (8)$$

$$\% \text{ unmet exclusion criteria} = \frac{|\{E(e_y) = \text{not excluded} \mid y = 1, 2, \ldots, N\}|}{N'} \quad (9)$$

$$\% \text{ not enough information} = \frac{|\{E(e_y) = \text{not enough information} \mid y = 1, 2, \ldots, N\}|}{N'} \quad (10)$$

$$N' = N - |\{E(e_y) = \text{not applicable} \mid y = 1, 2, \ldots, N\}| \quad (11)$$

LLM aggregation: In addition, TrialGPT also uses LLMs to aggregate the criterion-level predictions by the prompt shown in Supplementary Table 5. The generated scores are directly used as aggregation scores. Specifically, we consider two main features: general relevance and eligibility: The general relevance ($R$) indicates how relevant a patient is to clinical trial, while the eligibility score ($S$) denotes how eligible the patient is to the clinical trial. We restrict that:

$$0 \leq R \leq 100 \quad (12)$$

where 0 indicates that the patient is irrelevant to the clinical trial and 100 suggests that the patient is exactly relevant to the clinical trial. We further restrict that:

$$-R \leq S \leq R \quad (13)$$

based on the assumptions that the absolute value of eligibility cannot be higher than relevance.

Feature combination: We further combine the linear and LLM aggregation features, generating the improved scores for ranking and

excluding:

$$combination = \% \, met \, inclusion \, criteria - \mathbb{I}(\% \, unmet \, inclusion \, criteria > 0) \\ - \mathbb{I}(\% \, met \, exclusion \, criteria > 0) + \% \, LLM \, general \, relevance \\ + \% \, LLM \, eligibility \, score$$

(14)

where $\mathbb{I}$ is an indicator function:

$$\mathbb{I}(condition) = \begin{cases} 1, \text{if condition is True} \\ 0, \text{if condition is False} \end{cases}$$

(15)

## Criterion-level expert evaluation

Three annotators (F.C., C.G., and C.F.) are provided with 1,015 pairs of patient-criterion predictions by TrialGPT sampled from our patient cohorts. For each patient-criterion pair, the annotators first evaluate the correctness of TrialGPT explanation by "Correct", "Partially Correct" or "Incorrect". If at least two annotators provide the same label, it will be used as the consensus. If annotators choose three different scores for a patient-criterion pair, it will be labeled as "Partially Correct". The annotators then annotate the relevant sentence locations for the criterion, and the consensus is the union of all annotator-provided sentences. Finally, the annotator provides the annotation of eligibility, with the same candidate label set as that of TrialGPT. Similarly, if at least two annotators assign the same eligibility label, it will be used as the consensus. If all three annotators assign different eligibility labels, a second round of discussion will be scheduled until there is a consensus label.

## Compared methods

The core of TrialGPT lies in the prediction of patient-criterion eligibility. These predictions are then aggregated for the ranking and excluding tasks. As such, we compare TrialGPT to a variety of pre-trained language models that can predict patient-criterion eligibility. Since there are no existing patient-criterion eligibility annotations for training a supervised model, we consider transfer learning from the biomedical NLI datasets. Specifically, we use three categories of baselines: dual-encoder, cross-encoder, and encoder-decoder.

Dual-encoder models are also known as bi-encoder, where the patient note and the criterion are separately encoded by pre-trained transformers, and the eligibility is modeled as the similarity of the encoding vectors:

$$score(ranking) = \frac{\sum_{x=1}^{M} V(patient)^T V(i_x)}{M} - \frac{\sum_{y=1}^{N} V(patient)^T V(e_y)}{N}$$

(16)

$$score(excluding) = \frac{\sum_{y=1}^{N} V(patient)^T V(e_y)}{N}$$

(17)

and:

$$V(patient) = Enc([s_1, s_2, \ldots, s_P]) \in \mathbb{R}^h$$

(18)

$$V(i_x) = Enc(i_x) \in \mathbb{R}^h$$

(19)

$$V(e_y) = Enc(e_y) \in \mathbb{R}^h$$

(20)

where Enc denotes the pre-trained transformer encoder, and $h$ is the dimension of the vector representations.

Cross-encoder models take both the patient note and the criterion as input, which enables cross-attention computations between the tokens in both texts. The eligibility prediction is modeled as a 3-way classification task based on the special [CLS] embedding of BERT[44]. We use label space mapping functions $f$ that maps an NLI label to an eligibility label:

$$E(i_x) = f_{inc}(CrossEnc([s_1, s_2, \ldots, s_P], i_x))$$

(21)

$$E(e_y) = f_{exc}(CrossEnc([s_1, s_2, \ldots, s_P], e_y))$$

(22)

where

$$f_{inc}(l) = \begin{cases} \text{included, if } l = \text{entailment} \\ \text{notincluded, if } l = \text{contradiction} \\ \text{no relevant information, if } l = \text{neutral} \end{cases}$$

(23)

and

$$f_{exc}(l) = \begin{cases} \text{excluded, if } l = \text{entailment} \\ \text{not excluded, if } l = \text{contradiction} \\ \text{no relevant information, if } l = \text{neutral} \end{cases}$$

(24)

Then we compute the combination scores based on the criterion-level prediction, similar to the feature combination strategy used by TrialGPT:

$$combination(ranking) = \% \, met \, inclusion \, criteria - \% \, unmet \, inclusion \, criteria \\ - \% \, met \, exclusion \, criteria + \% \, unmet \, exclusion \, criteria$$

(25)

$$combination(excluding) = \mathbb{I}(\% \, unmet \, inclusion \, criteria > 0) \\ + \mathbb{I}(\% \, met \, exclusion \, criteria > 0) - \% \, met \, inclusion \, criteria$$

(26)

Encoder-decoder models also take both the patient note and the criterion as input to the encoder, but instead of outputting a classification prediction, they generate the predicted NLI labels, e.g., "entailment", "contradiction", or "neutral". These NLI labels are then mapped to eligibility labels that will be aggregated into the combination scores by the same methods described above for cross-encoder models.

## Evaluation settings

Following the SIGIR and TREC evaluation guidelines, we define the relevance score $r_i$ of a clinical trial $c_i$ given a specific patient as:

$$r_i = \begin{cases} 0, \text{if } E(c_i) \in \{\text{irrelevant, unlabeled}\} \\ 1, \text{if } E(c_i) \in \{\text{ineligible, potential}\} \\ 2, \text{if } E(c_i) = \text{eligible} \end{cases}$$

(27)

We report the recall at different depths $k$ for the first stage retrieval. Specifically, we denote the retrieved list of clinical trials as $[c_1, c_2, \ldots, c_k]$, where $k$ is the number of considered candidates. Their relevance scores are denoted as $[r_1, r_2, \ldots, r_k]$. Recall@k is a measurement of retrieval quality, which is computed by:

$$Recall@k = \frac{\sum_{x=1}^{k} r_x}{R}$$

(28)

where R is the sum of all clinical trial relevance score in the considered collection.

We report NDCG@10 and P@10 for ranking candidate clinical trials, and AUROC for excluding ineligible clinical trials. Patient-trial pairs with unlabeled eligibility are not included for the computation of metrics.

For computing NDCG@10 and P@10, we denote the ranked list of clinical trials as $[c_1, c_2, \ldots, c_k]$, where $k$ is the number of considered candidates. Their relevance scores are denoted as $[r_1, r_2, \ldots, r_k]$. NDCG@k is a measurement ranking quality, which is computed by:

$$\text{NDCG}@k = \frac{\text{DCG}@k}{\text{IDCG}@l} \quad (29)$$

where

$$\text{DCG}@k = \sum_{x=1}^{k} \frac{r_x}{\log_2(i+1)} \quad (30)$$

and

$$\text{IDCG}@k = \sum_{x=1}^{k} \frac{r'_x}{\log_2(i+1)} \quad (31)$$

where $[r'_1, r'_2, \ldots, r'_T]$ denotes the relevance of an ideal ranking.

P@10 is another metric for ranking quality, computed by:

$$\text{P}@k = \frac{\sum_{x=1}^{k} r_x}{\max(\mathcal{R}) \times k} \quad (32)$$

where $\mathcal{R}$ denotes the set of relevance labels.

We draw the receiver operating characteristic (ROC) curve and compute the area under the ROC curve (AUROC) values using the sklearn package in Python.

## Pilot user study

The pilot user study mimics a common task at the cancer center, where the trial organizer screens patient referral requests against the candidate clinical trials. The patient note varies from a short paragraph in the email to a long document sent via fax. We consider six clinical trials (NCT04432597, NCT05012098, NCT04287868, NCT04847466, NCT04719988, and NCT04894370) as the candidates, where the first four are managed by one physician co-author (C.F.) and the other two are highly related. The physician also created six clinical vignettes (3 short and 3 long) based on real patient encounters, with the personally identifiable information modified. The task objective is to screen whether the patient is definitely ineligible ("No") or potentially eligible and should be included for further investigation ("Maybe"). Two MD annotators (Q.J. and E.X.) recorded the time needed to make the decision after familiarizing themselves with the patient's note. Each annotator screens half of the patient-trial pairs with TrialGPT and another half without. We also ensure that each patient-trial pair is screened by one annotator with TrialGPT and another without. The evaluation setting is visualized in Fig. 5a.

## Reporting summary

Further information on research design is available in the Nature Portfolio Reporting Summary linked to this article.

## Data availability

The TREC Clinical Trial 2021 and 2022 cohorts can be downloaded from http://www.trec-cds.org/2021.html and http://www.trec-cds.org/2022.html, respectively. The SIGIR cohort is publicly available at https://data.csiro.au/collection/csiro:17152. The clinical vignettes for the user study are available in the Supplementary Materials. The criterion-level annotations generated in this study have been deposited in the Hugging Face database under accession code: https://huggingface.co/datasets/ncbi/TrialGPT-Criterion-Annotations. Preprocessed data files generated in this study have been deposited in the GitHub database under accession code: https://github.com/ncbi-nlp/TrialGPT [45]. Source data are provided with this paper.

## Code availability

TrialGPT is publicly available at https://github.com/ncbi-nlp/TrialGPT [45].

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

## Acknowledgements

This research was supported by the NIH Intramural Research Program, National Library of Medicine. We would like to thank the organizers of the SIGIR and TREC Clinical Trial tasks for making their datasets publicly available.

## Author contributions

Q.J., Z.W., C.F., J.S., and Z.L. designed the study. Q.J. conducted the data collection, model construction, model evaluation, and manuscript drafting. Z.W. and Y.Y. carried out the data collection and analysis. Q.J., C.F., F.C., C.G., D.B., and E.X. contributed to the data annotation. Z.L. supervised the study. All authors contributed to writing the manuscript and approved the submitted version.

## Competing interests

The authors declare no competing interests.
