## [Peer Review File · Nature Communications]

Reviewers' Comments:

Reviewer #1:

Remarks to the Author:

This paper investigates using LLMs for evaluating patient eligibility for clinical trial criteria. This is generally an interesting direction, but the paper hasn't demonstrated sufficient justification for its novelty or significance.

First, the paper proposed TrialGPT as a novel method for clinical trial matching. However, upon close inspection, it appears that the authors were just using GPT4. The name "TrialGPT" is thus very misleading. It would imply significant adaptation (e.g., fine-tuning GPT-X for clinical trial domains), which is not really what happened. The authors suggested that what they propose is actually a general framework, but what's repeatable is actually really straightforward, essentially how clinical trial matching is reduced to first criteria-level matching, then trial-level aggregation. The key aspects like prompt design etc. are all specific to GPT4, and there is no evidence about their generalizing to arbitrary LLMs. Therefore, it doesn't seem justifiable at all to present the study as if there is really a novel model.

Given that GPT4 is the central LLM, the paper should offer more details on prompt engineering and other related aspects. There are a plethora of studies evaluating GPT4 for related explorations such as clinical trials, entailment, prompting strategies in biomedical domain. But the paper contains little discussion about related work. E.g., the MedPrompt paper by Nori et al. ("Can Generalist Foundation Models Outcompete Special-Purpose Tuning? Case Study in Medicine") conducts detailed studies on various prompting strategies, including ones that were used in this exploration. The authors should discuss such related work, and if/how their work is novel compared to prior work.

The authors should also discuss the significance of their problem formulation and its pros and cons. E.g., a prevailing alternative approach, as adopted by some methods cited in the paper, would proceed by structuring both patient information and eligibility criteria, thus reducing matching to scalable database query. By contrast, it seems that the proposed method has to run LLMs on each patient-criterion pair, which will grow as the cross product of patient number and trial number. This seems not very scalable, esp. compared to the alternative approach that only scales linearly in patient number + trial number. Indeed, the paper stated that "For each patient, we sample at most 50 clinical trials for each eligibility category". In practice, this would mean that an additional retrieval step is required, and it's unclear how it would impact the precision / recall tradeoff. The authors should discuss such ramifications and how they might impact adoption.

The paper compares with a number of prior generation models (BERT models), which seem weak baselines. It seems that they were used as is, essentially as entailment classifier. Does this mean the difference simply stems from the fact that GPT4 is a better entailment classifier (which is well known)? The study does not seem specific to clinical trial and it's unclear what useful lessons could we draw from this exploration for clinical research.

Reviewer #2:

Remarks to the Author:

-- Summary

Jin et al. proposed and evaluated a large language model (LLM) framework, named TrialGPT, for matching patients to clinical trials based on patient notes and inclusion-exclusion criteria. TrialGPT predicts a patient's relevancy and eligibility for a trial on a criterion level, along with an explanation of the prediction and the locations of the supporting sentences in the patient note. In a cohort of 183 patients, TrialGPT achieved a criterion-level accuracy of 87.3%, close to the performance of three physicians. The trial-level scores derived from criterion-level predictions correlated with the ground-truth eligibility labels and outperformed various pre-trained language models in ranking and excluding trials. In a small pilot user study, TrialGPT significantly reduced the screening time needed for clinicians.

-- Significance & Originality

This study is novel in using LLM to generate criterion-level eligibility prediction, which enables explainability meaningful for clinical use and results in high trial-level ranking and exclusion performance. The user study further showcases the framework's utility for assisting patient-trial matching. Overall, the study is significant in its novel approach and clinical implications.

-- Comments

1. How were the 1050 patient-criterion pairs selected? There are 183 patients in the cohort, so on average, 5.7 patient-criterion pairs are evaluated per patient, which seems to be one trial per patient.
2. Consider providing confidence intervals to the results in "Evaluating relevant sentence locations" (line 156-162; Figure 2b; the accuracy in Figure 2d-c)
3. Could you clarify what line 220 means, "unlike other graphs that are either monotonically increasing or decreasing with regard to the irrelevant-ineligible-eligible

order”. I see the monotonicity in Figure 3(a), (b), and (g) in terms of the median, but not in the rest of the subplots. I notice the median line “missing” in some of the boxes, and I guess they overlap with the edges. Maybe using a different color for the median line will help explain the monotonicity better?

4. Could you briefly explain the selection of SciFive, BioBERT, PubMedBERT, SapBERT, BioLinkBERT for the comparison of ranking and excluding trials? Are they (or their variations) the state-of-the-art methods in patient-to-trial benchmarks?

5. Line 321: There is a difference between the variations of screening time with and without TrialGPT within the annotator, should you use Welch's t-test?

6. Line 322: I am not sure if paired t-test is the right test to use, given that “...for each patient-trial pair, one annotator screened with TrialGPT, and another without”. I think using a paired t-test would be appropriate if you were comparing the screening time of the same doctor (group of doctors) with and without TrialGPT.

-- Minor Comments

1. Consider adding the following for references: Zhuang, Shengyao, Bevan Koopman, and Guido Zuccon. "Team IELAB at TREC Clinical Trial Track 2023: Enhancing Clinical Trial Retrieval with Neural Rankers and Large Language Models." arXiv preprint arXiv:2401.01566 (2024).

2. Consider defining “M” and “N” in line 451 where they first occur

Reviewer #3:

Remarks to the Author:

This manuscript presents a GPT-4 based method for clinical trial matching. The authors propose a prompting method to use GPT for identifying whether a patient description matches the inclusion and exclusion criteria of a clinical trial description. Additional experiments were performed using criterion-level annotations. GPT-4 was compared against a wide variety of BERT-based baselines, showing superior results.

Overall, this is an excellent and well-written paper. I really appreciated reading this and the work the authors have done. On the other hand, several choices made by the authors appear to be unnecessary deviations from standard practice that they require a justifiable explanation as well as more details:

1. "we sample at most 50 clinical trials for each eligibility category" -- why? In practice this means that you are removing non-relevant trials from the evaluation and preventing comparison to existing systems (e.g., Koopman et al.'s and the TREC participants). If the argument is the cost of running GPT-4, then I would argue that this is a MAJOR limitation of using GPT for this task. CT.gov has something like 500k trials (which is also relevant for the next point below).

2. There is no evaluation of the information retrieval (IR) component of this task. The authors even obliquely acknowledge this ("Several pilot studies have also explored using LLMs to enhance the first-stage retrieval") but almost cast it as a strength, not a weakness of the paper. Again, with 500k clinical trials in CT.gov some kind of IR system is going to be necessary. There is not reason to believe, and actually evidence from IR to disbelieve, that an LLM's performance is independent of an initial search engine's performance -- it could very well be that precisely the "hard" cases the LLM does so well on compared to other models that are also the cases not retrieved by the IR system. TREC uses pooled results, so just because ONE retrieval system pulled it back (out of dozens), doesn't mean most would.

3. What datasets were the 1,015 patient-criterion pairs drawn from?

4. Where are the candidate clinical trials in the ranking analysis drawn from? My assumption is that it is just the ~300 trials that were already sampled. I have a significant problem with IR metrics like NDCG and P@10 being used when non-IR methods are used to select the candidates. A major issue with this paper is that the many systems that have been published on these three datasets are not comparable. When one compounds this with metrics that give the appearance of comparability that creates a major point of confusion (e.g., someone might compare the numbers in Table 2 to some paper published at SIGIR and conclude TrialGPT is better, when in fact the numbers cannot be compared). LLMs generate a lot of hype, and it is scientifically appropriate to ensure information is present that makes it clear what can and cannot be legitimately compared.

5. This is not "a systematic evaluation of LLMs", as the Discussion says. It is a comparison of a single LLM to many BERT models. It evaluates TrialGPT across several important dimensions of performance. But it does not systematically evaluate LLMs.

6. The authors claim "Our work supports the position that AI models for clinical trial matching should not be designed to replace human recruiters but to empower them...". Yet, nothing in this manuscript supports this claim. The authors are correct to point out that fully-automated trial matching is not appropriate, but I don't think anyone reading this paper is going to assume that if TrialGPT flags a patient then they will automatically

be enrolled. So if the authors believe that one of their experiments actually supports this claim, then by all means justify the statement. Otherwise, it would be best to formulate this opinion as a counterfactual (e.g., "Our work does not justify the position that clinical trial matching should be fully automatic and exclude human recruiters...", but phrase is however you wish).

7. Finally, I feel that the related work here is not particularly well-cited. There's been a lot of work in this space recently. I realize this manuscript started as a July 2023 preprint, but there has been a ton of work since then and I only noticed four additional citations related to this area (two of which are self-cites). So the authors need to do better at identifying relevant work and describing the particular gap that this paper fills. Here's an incomplete list:

<https://arxiv.org/abs/2402.05125>

<https://pubmed.ncbi.nlm.nih.gov/37952206/>

<https://arxiv.org/abs/2401.01566> (not to mention the other TREC participants, for which LLMs were highly utilized)

<https://arxiv.org/abs/2312.09958>

This is only a cursory search, of course. This area has consistently been mentioned as one of the biomedical areas that LLMs can make the most immediate impact, so papers are coming out thick and fast and it is important for a journal like this to make the case for why this approach is impactful.

Minor:

- Abstract:

-- "first-of-its-kind" is a difficult to assess claim, especially since we don't have good ways of defining the official definition of a "large" language model. But, regardless, some of the TREC CT 2023 participants used LLMs and that was back in August of 2023, so probably the best thing to do is to remove this term.

-- "184 patients" -> "184 synthetic patients"

- Introduction:

-- Citation #6 looks wrong.

-- "184 patients" -> "184 synthetic patients"

-- Figure 1c - it isn't clear what the little pictures are indicative of... it gives the sense that TrialGPT is explicitly considering the patient's genes, pill meds, injected meds, and microbiology cultures. Of course TrialGPT doesn't do this, but point is that it is strange to understand.

- Results:

-- Only two citations for three datasets (appears that TREC CT 2022 isn't cited). Properly citing scientific data is a major focus at the moment.

-- "Interestingly, patients meet more exclusion..." -> "Importantly, patients meet more exclusion" (it isn't interesting, it is a well-known and obvious fact to anyone that

understands clinical trials, but it is important and worth the authors pointing out)

- Discussion:

-- "We also notice... might be over-simplified... (geolocations, recruitment status, etc.)"

-- It is of course appropriate to point out the aspects of clinical trial search that are not covered by the datasets utilized, but this wording suggests the authors have noticed something that isn't fully explicit from the dataset creators. The dataset creators do in fact make this explicit and (in their minds) exclude geolocation/recruitment status with good reason. Also, "etc." implies there are likely other things, but the dataset creators only explicitly excluded geolocation and recruitment status from consideration, so if the authors have noticed additional issues not mentioned by the dataset creators, please be explicit about this. Otherwise it would be more appropriate to phrase this sentence along the following lines: "Further, the SIGIR and TREC datasets focus on the semantics of the clinical trial inclusion/exclusion criteria, excluding factors such as geolocation and trial recruitment status as these are addressable through traditional structured query approaches. As a result, any use of TrialGPT would similarly need to ensure the identified trials are appropriate for a patient along these lines as well."

-- "GPT-4 is the most capable" -> "GPT-4 is currently the most capable"

-- "our pilot user study is of limited scope in sample size" -> state the sample size, e.g., "(n=XX)".

DEPARTMENT OF HEALTH & HUMAN SERVICES

Public Health Service

National Institutes of Health
National Library of Medicine
Bethesda, Maryland 20894

June 3, 2024

Dear Reviewers,

We sincerely thank you for your thorough evaluation and insightful comments on our manuscript titled, “TrialGPT: Matching Patients to Clinical Trials with Large Language Models”. We highly appreciate the time and effort you invested in reviewing our work, and the constructive feedback provided. The comments have been vital in enhancing the quality and clarity of our manuscript. We have carefully addressed each comment and suggestion in a systematic manner and believe that the revisions have significantly improved the manuscript. In the following sections, we provide point-by-point responses to each of the comments, alongside the corresponding changes made in the manuscript.

Response to Reviewer #1

Overall Comment

This paper investigates using LLMs for evaluating patient eligibility for clinical trial criteria. This is generally an interesting direction, but the paper hasn't demonstrated sufficient justification for its novelty or significance.

Response

We are grateful for your constructive feedback on our manuscript. We have provided point-by-point responses to your concerns in the following sections.

Comment 1

First, the paper proposed TrialGPT as a novel method for clinical trial matching. However, upon close inspection, it appears that the authors were just using GPT4. The name "TrialGPT" is thus very misleading. It would imply significant adaptation (e.g., fine-tuning GPT-X for clinical trial domains), which is not really what happened. The authors suggested that what they propose is actually a general framework, but what's repeatable is actually really straightforward, essentially how clinical trial matching is reduced to first criteria-level matching, then trial-level aggregation. The key aspects like prompt design etc. are all specific to GPT4, and there is no evidence about their generalizing to arbitrary LLMs. Therefore, it doesn't seem justifiable at all to present the study as if there is really a novel model.

Response

Thank you for your comments. In the revision, we have clarified that **TrialGPT is a framework for assisting in three stages of the patient-trial matching task via zero-shot learning with off-the-shelf large language models**, instead of a new fine-tuned model.

In addition to the previous criterion-level matching (now as the TrialGPT-Matching component) and trial-level aggregation (now as the TrialGPT-Ranking component), we have also introduced TrialGPT-Retrieval as the first step, making the TrialGPT an end-to-end solution. GPT-4 was selected as the backbone in the original submission as the most capable LLM. This is out of the same motivation as the MedPrompt paper which also uses GPT-4 as the main LLM. In this revision, we further added TrialGPT results using GPT-3.5. Nonetheless, we have acknowledged the potential limitations of using GPT series in the section “*Future studies should explore using and fine-tuning other open-source LLMs as alternatives*”.

Although well-crafted prompting is one of the key elements that make TrialGPT succeed, this study's focus is on exploring how to adapt LLMs to streamline patient-trial matching in a scalable and explainable paradigm. In particular, we compared TrialGPT to human clinicians at various stages: (1) TrialGPT-Retrieval generates better keywords than clinicians for first-stage trial retrieval; (2) TrialGPT-Matching performs similarly to clinicians at criterion-level eligibility prediction; and (3) TrialGPT can save up to 42.6% of screening time of clinical trials. Additionally, we also compared TrialGPT-Ranking with BERT-based entailment models and

the results confirmed its superiority. That said, we have also acknowledged the need for in-depth analyses of different prompts for the TrialGPT framework as a future direction in the revision: *“Second, while our study proposes a novel framework of patient-to-trial matching with LLMs, there are various other prompting strategies for each of the TrialGPT component that are worth exploration in future work”*.

Finally, we would like to clarify that using “GPT” in the method name does not necessarily imply fine-tuning. For example, a recent paper entitled *“Assessing GPT-4 for cell type annotation in single-cell RNA-seq analysis”* developed a package called GPTCelltype, which uses off-the-shelf GPT-4 for cell type annotation (Hou & Ji, *Nature Methods*, 2024). Nonetheless, we removed TrialGPT from the title but elect to keep it in the article.

Comment 2

Given that GPT4 is the central LLM, the paper should offer more details on prompt engineering and other related aspects. There are a plethora of studies evaluating GPT4 for related explorations such as clinical trials, entailment, prompting strategies in biomedical domain. But the paper contains little discussion about related work. E.g., the MedPrompt paper by Nori et al. (“Can Generalist Foundation Models Outcompete Special-Purpose Tuning? Case Study in Medicine”) conducts detailed studies on various prompting strategies, including ones that were used in this exploration. The authors should discuss such related work, and if/how their work is novel compared to prior work.

Response

Thank you for your suggestions. In the original manuscript, we listed various capabilities and evaluations of GPT-4 in the Introduction section, including both general capabilities in medicine and the ones specific to clinical trials. In this revision, we have added the citation MedPrompt work as one example of LLMs' question-answering capability.

However, TrialGPT is different from MedPrompt in two major ways:

(1) MedPrompt tackles a different task: multi-choice medical question answering, and the main contribution is to test and combine existing prompting strategies, e.g., chain-of-thought, few-shot prompting, and ensemble, to obtain the best performance. By contrast, this work focuses on patient-trial matching tasks, including Retrieval, Matching, and Ranking. The main contribution of our study is to propose such an end-to-end pipeline for patient-to-trial matching with large language models, with rigorous evaluations at each step;

(2) MedPrompt adopts various prompting strategies, including using the training data of the QA datasets to provide kNN-based few-shot exemplars for in-context learning. By contrast, TrialGPT needs to resolve a challenge where no prior criterion-level eligibility annotations are available for training. As such, it emphasizes how to adapt LLMs for patient-trial matching in zero-shot learning.

As mentioned in our previous response, although well-crafted prompting is one of the key elements that make TrialGPT succeed, this study's focus is on exploring how to adapt LLMs to streamline patient-trial matching in a scalable and explainable paradigm. In particular, we

compared TrialGPT to human clinicians at various stages: For example, TrialGPT-Retrieval generates better keywords than clinicians for first-stage trial retrieval; TrialGPT-Matching performs similarly to clinicians at criterion-level eligibility prediction; and TrialGPT can save up to 42.6% of screening time of clinical trials. Additionally, we also compared TrialGPT-Ranking with zero-shot BERT-based entailment models and the results confirmed its superiority.

In addition to discussing MedPrompt in the manuscript, we have also acknowledged the need for in-depth analyses of different prompts for the TrialGPT framework as a future direction in the revision, per this comment.

Comment 3

The authors should also discuss the significance of their problem formulation and its pros and cons. E.g., a prevailing alternative approach, as adopted by some methods cited in the paper, would proceed by structuring both patient information and eligibility criteria, thus reducing matching to scalable database query.

Response

Thank you for your thoughtful comments. In the revision, we have highlighted the benefits of TrialGPT for three important tasks in patient-trial matching: we improve its scalability by the newly introduced TrialGPT-Retrieval component that filters out irrelevant clinical trials at the initial stage; TrialGPT-Matching provides accurate criterion-level predictions with faithful

explanations; and TrialGPT-Ranking can aggregate the TrialGPT-Matching results into trial-level scores for ranking clinical trials.

In contrast, the alternative approaches mainly consider one of the tasks or approach it in a less scalable and flexible way. For example, criteria2query and Wong *et al.* (2023) requires the patient notes and clinical trial criteria to be accurately structured into the same vocabulary space, which can often be challenging due to the various writing styles of the clinical trial criteria. As Wong *et al.* (2023) have pointed out in their paper: *“In particular, it is not always possible to map an extracted criteria into an existing concept or ontology; and indeed, we found quite a few examples in our study where this was not possible. There are also various subtleties in the criteria language that are difficult to capture completely into a logical formula”*. In TrialGPT, the criterion-level matching is achieved by LLMs reading both the patient summary and the criteria in natural language, which does not require the criteria to follow any formats and is thus more flexible. We have added the discussion of pros and cons of our approach against the others in the revision.

Comment 4

By contrast, it seems that the proposed method has to run LLMs on each patient-criterion pair, which will grow as the cross product of patient number and trial number. This seems not very scalable, esp. compared to the alternative approach that only scales linearly in patient number + trial number. Indeed, the paper stated that "For each patient, we sample

at most 50 clinical trials for each eligibility category". In practice, this would mean that an additional retrieval step is required, and it's unclear how it would impact the precision / recall tradeoff. The authors should discuss such ramifications and how they might impact adoption.

Response

Thank you for your constructive comment, and we agree that a retrieval step can add significant computational benefit to the task. Therefore, we have added a new component TrialGPT-Retrieval in the revision to make TrialGPT an end-to-end framework for clinical trial matching with large language models. Specifically, given a patient summary, TrialGPT-Retrieval can generate a list of keywords and uses a hybrid-fusion retrieval mechanism to return the top hundreds to thousands of relevant clinical trials from the initial collection. We added a new section titled "*TrialGPT-Retrieval can generate keywords for effective clinical trial filtering*" to show the evaluation results of TrialGPT-Retrieval. Overall, TrialGPT-Retrieval can recall over 90% of the relevant clinical trials using only less than 6% of the initial collection. Keywords generated by TrialGPT-Retrieval not only surpass the raw note for retrieval performance, but also outperform the clinician-generated keywords in the SIGIR cohort.

Comment 5

The paper compares with a number of prior generation models (BERT models), which seem weak baselines. It seems that they were used as is, essentially as entailment classifier. Does this mean the difference simply stems from the fact that GPT4 is a better entailment classifier (which is well known)? The study does not seem specific to clinical trial and it's unclear what useful lessons could we draw from this exploration for clinical research.

Response

In the revised manuscript, there are several main comparisons showing the superiority of TrialGPT at different stages of patient-to-trial matching:

(1) For retrieval, we show that LLM-generated keywords are better than both clinician-generate keywords and the raw note for the first-stage retrieval of relevant clinical trials from a large collection;

(2) For matching, we show that TrialGPT-Matching is close to clinician performance for eligibility prediction, and the model can also provide faithful explanations in natural language and accurate locations of relevant sentences in the raw patient notes;

(3) For ranking, we compare TrialGPT with entailment models based on BERT and T5. We would like to clarify that there is no open and large-scale patient-trial matching dataset for training, hence we mainly focus on the zero-shot scenario. The selected baselines are the cutting-edge natural language entailment prediction models pre-trained on various natural language inference tasks (MNLI, SNLI, STSB, MedNLI, SciNLI, and SciTail). They can be considered as the best zero-shot baselines;

(4) We also compare the screening time with and without TrialGPT to show the practical value that our method can bring to assist clinicians in streamlining patient recruitment.

Despite LLMs like GPT-4 hold the strong capability for general natural language understanding and generating, how to translate them to clinical trial matching and ranking performance has remained elusive. In addition, our experiment setups are close to the practice in clinical trial patient recruitment. The results, hence, manifest as practical guidelines for applying and improving LLM-based systems for patient recruitment in the future.

Responses to Reviewer #2

Comment (Summary, Significance, and Originality)

Jin et al. proposed and evaluated a large language model (LLM) framework, named TrialGPT, for matching patients to clinical trials based on patient notes and inclusion-exclusion criteria. TrialGPT predicts a patient's relevancy and eligibility for a trial on a criterion level, along with an explanation of the prediction and the locations of the supporting sentences in the patient note. In a cohort of 183 patients, TrialGPT achieved a criterion-level accuracy of 87.3%, close to the performance of three physicians. The trial-level scores derived from criterion-level predictions correlated with the ground-truth eligibility labels and outperformed various pre-trained language models in ranking and excluding trials. In a small pilot user study, TrialGPT significantly reduced the screening time needed for clinicians.

This study is novel in using LLM to generate criterion-level eligibility prediction, which enables explainability meaningful for clinical use and results in high trial-level ranking and exclusion performance. The user study further showcases the framework's utility for assisting patient-trial matching. Overall, the study is significant in its novel approach and clinical implications.

Response

We are grateful for your positive remarks. We have provided point-by-point responses to your other comments below.

Major Comment 1

How were the 1050 patient-criterion pairs selected? There are 183 patients in the cohort, so on average, 5.7 patient-criterion pairs are evaluated per patient, which seems to be one trial per patient.

Response

They are sampled from 105 patient-trial pairs from 53 patients in the SIGIR cohort. In the revision, we have added the clarification to the results section. The reasons for choosing the SIGIR cohort for this purpose are two-fold:

(1) We have released the GPT-4 annotations and human judgments on these patient-criterion pairs at <https://huggingface.co/datasets/ncbi/TrialGPT-Criterion-Annotations>. Keeping the annotations only to the single SIGIR cohort makes sure that researchers can use the two TREC cohorts as independent test sets for robust evaluations.

(2) We found that the SIGIR cohorts are also annotated with clinician-generated keywords. Therefore, we have compared TrialGPT-Retrieval to them for the first-stage retrieval performance. As such, it keeps the results comparable by annotating the same cohort and comparing the TrialGPT-Matching to clinician eligibility labels for the same patients.

Major Comment 2

Consider providing confidence intervals to the results in “Evaluating relevant sentence locations” (line 156-162; Figure 2b; the accuracy in Figure 2d-c)

Response

Thank you for your suggestion. We have added 95% confidence interval to all numbers in Figures 2b, 2c, and 2d. The confidence intervals are estimated by bootstrapping with 10,000 times of resampling.

Major Comment 3

Could you clarify what line 220 means, “unlike other graphs that are either monotonically increasing or decreasing with regard to the irrelevant-ineligible-eligible order”. I see the monotonicity in Figure 3(a), (b), and (g) in terms of the median, but not in the rest of the subplots. I notice the median line “missing” in some of the boxes, and I guess they overlap with the edges. Maybe using a different color for the median line will help explain the monotonicity better?

Response

We intended to highlight that ineligible patient-trial pairs meet the most exclusion criteria. This insight can be exploited to develop criterion-level prediction aggregation methods for trial-level eligibility ranking. In the revision, we have removed this description because we have added more label groups to the patient-trial pairs, which now include: unlabeled,

irrelevant (label 0 in both SIGIR and TREC cohorts), excluded (label 1 in the TREC cohorts), potential (label 1 in the SIGIR cohort), and eligible (label 2 in both SIGIR and TREC cohorts).

It is correct that the median lines sometimes overlap with the edges in the box plots, which means that the 50% percentile value and the 25%/75% percentile value(s) are the same. It usually happens when most of the aggregated values are 0, such as the proportion of met exclusion criteria and the proportion of unmet inclusion criteria for eligible patient-trial pairs. As we dropped the monotonicity claims in the revision, we keep using the standard way to draw the box plots.

Major Comment 4

Could you briefly explain the selection of SciFive, BioBERT, PubMedBERT, SapBERT, BioLinkBERT for the comparison of ranking and excluding trials? Are they (or their variations) the state-of-the-art methods in patient-to-trial benchmarks?

Response

To compare with TrialGPT-Ranking, we have selected three representative classes of language models for eligibility prediction: dual-encoders (BioBERT, PubMedBERT, SapBERT), encoder-decoders (SciFive), and a cross-encoder (BioLinkBERT). We would like to clarify that there is no open and large-scale patient-trial matching dataset for training, hence we mainly focus on the zero-shot scenario. The selected baselines are the cutting-edge natural language entailment prediction models pre-trained on various natural language inference

tasks (MNLI, SNLI, STSB, MedNLI, SciNLI, and SciTail). They can be considered as the best zero-shot baselines.

Major Comment 5

Line 321: There is a difference between the variations of screening time with and without TrialGPT within the annotator, should you use Welch's t-test? Line 322: I am not sure if paired t-test is the right test to use, given that "...for each patient-trial pair, one annotator screened with TrialGPT, and another without". I think using a paired t-test would be appropriate if you were comparing the screening time of the same doctor (group of doctors) with and without TrialGPT.

Response

Thank you for your comment. In practice, comparing the annotation time for the same patient-trial pair by the same physician under different conditions—with and without TrialGPT—is not feasible. Conducting annotations under one setting first inevitably increases the physician's familiarity with the data. This prior exposure can bias the results, rendering any subsequent comparison between the two settings unfair. As such, we designed our user study to have the interleaving annotation pattern, where each patient-trial pair is annotated by one doctor with TrialGPT and another doctor without TrialGPT. In addition, for each doctor, we ensure that half of the annotations are done with TrialGPT and another half without TrialGPT, which is also true at each individual case- and trial-level. In the revision, we show that there is no significant time difference between Annotator X and

Annotator Y in screening with TrialGPT (32.7s v.s. 37.8s, $p=0.73$) or without TrialGPT (57.5s v.s. 65.4s, $p=0.75$), which indicates that they have similar screening capabilities and can be considered as “the same group of doctors”. In the revision, we have added such justification of using paired t-test in the manuscript.

Minor Comment 1

Consider adding the following for references: Zhuang, Shengyao, Bevan Koopman, and Guido Zuccon. "Team IELAB at TREC Clinical Trial Track 2023: Enhancing Clinical Trial Retrieval with Neural Rankers and Large Language Models." arXiv preprint arXiv:2401.01566 (2024).

Response

Thank you for pointing out this relevant paper. In the revision, we have cited this together with “*Large language models for healthcare data augmentation: An example on patient-trial matching*” as demonstrations of data augmentation with synthetic data generated by large language models.

Minor Comment 2

Consider defining “M” and “N” in line 451 where they first occur

Response

In the original manuscript, we defined “M” and “N” in lines 414 and 415, which appeared at the beginning of the TrialGPT method section. In the revision, we have moved the definition to their first occurrence in the Online Methods for TrialGPT-Matching.

Responses to Reviewer #3

Overall Comment

This manuscript presents a GPT-4 based method for clinical trial matching. The authors propose a prompting method to use GPT for identifying whether a patient description matches the inclusion and exclusion criteria of a clinical trial description. Additional experiments were performed using criterion-level annotations. GPT-4 was compared against a wide variety of BERT-based baselines, showing superior results.

Overall, this is an excellent and well-written paper. I really appreciated reading this and the work the authors have done. On the other hand, several choices made by the authors appear to be unnecessary deviations from standard practice that they require a justifiable explanation as well as more details.

Response

We appreciate your positive remarks and thoughtful feedback for our study. Please find our point-by-point responses to your concerns below.

Major Comment 1

"we sample at most 50 clinical trials for each eligibility category" -- why? In practice this means that you are removing non-relevant trials from the evaluation and preventing comparison to existing systems (e.g., Koopman et al.'s and the TREC participants). If the

argument is the cost of running GPT-4, then I would argue that this is a MAJOR limitation of using GPT for this task. CT.gov has something like 500k trials (which is also relevant for the next point below).

Response

Thank you for this thoughtful comment. In the revision, we have added the information retrieval (IR) component, **TrialGPT-Retrieval**, to the TrialGPT framework. Please see our response to your next comment for more details on this IR component. For each patient, we first use TrialGPT-Retrieval to get the top 500 candidate clinical trials from the initial collection. Then, we use TrialGPT-Matching and TrialGPT-Ranking to perform fine-grained reranking of the trials returned by TrialGPT-Retrieval. As such, the revised TrialGPT is an end-to-end framework and **no longer requires such a sampling process** mentioned in the comment.

Major Comment 2

There is no evaluation of the information retrieval (IR) component of this task. The authors even obliquely acknowledge this ("Several pilot studies have also explored using LLMs to enhance the first-stage retrieval") but almost cast it as a strength, not a weakness of the paper. Again, with 500k clinical trials in CT.gov some kind of IR system is going to be necessary. There is not reason to believe, and actually evidence from IR to disbelieve, that an LLM's performance is independent of an initial search engine's performance -- it could

very well be that precisely the "hard" cases the LLM does so well on compared to other models that are also the cases not retrieved by the IR system. TREC uses pooled results, so just because ONE retrieval system pulled it back (out of dozens), doesn't mean most would.

Response

Thank you for your constructive comment. We agree with you that an IR component is necessary for the task. Therefore, we have added a new component, TrialGPT-Retrieval, in the revision to make TrialGPT support the entire clinical trial matching workflow with large language models. Specifically, given a patient summary, TrialGPT-Retrieval generates a list of keywords, which are used to retrieve the top hundreds to thousands of relevant clinical trials from the initial collection. Our newly added evaluation results show that TrialGPT-Retrieval's keywords surpass the raw note for retrieval performance and outperform the clinician-generated keywords in the SIGIR cohort.

For the retrieval experiments, instead of starting with the entire clinicaltrials.gov with over 500 thousand listed clinical trials, we use the combination of all annotated clinical trials in the corresponding cohorts as the initial collection for two reasons:

(1) While these competition datasets provide valuable annotations for evaluating different clinical trial matching systems, the relevance judgments are inevitably biased toward the participating systems. As you have thoughtfully pointed out, TREC uses pooled evaluation, which means that only clinical trials retrieved by at least one participating

system are judged and the rest of clinical trials are automatically considered as irrelevant. In other words, there are False Negatives, which are potentially relevant or eligible clinical trials considered irrelevant simply because they have not been judged in the competition. Therefore, we combine all annotated clinical trials in the corresponding cohorts as the initial document collections to minimize such biases.

(2) We firmly believe that retrieving from all listed clinical trials in CT.gov is not necessary when using TrialGPT as intended in practical applications. Although there are nearly 500,000 clinical trials registered in clinicaltrial.gov, in a real-world setting, patients typically focus on active clinical trials in a specific geolocation, as proximity is an important factor for patients when considering participation in clinical trials. For instance, the largest number of active trials in a country is around 23,000 (i.e., the United States). The total annotated trials in the TREC 2021 and TREC 2022 cohorts are 26,149 and 26,581, respectively. We demonstrate TrialGPT-Retrieval is capable of locating a small subset of relevant trials from tens of thousands of trials in these initial corpora. Although the SIGIR cohort has fewer annotated trials, we compare TrialGPT-Retrieval to the clinician-generated keywords with the same set of initial trials and demonstrate its superior performance.

Based on this comment, we have added these considerations in the revision.

Major Comment 3

What datasets were the 1,015 patient-criterion pairs drawn from?

Response

These pairs are sampled from 105 patient-trial pairs from 53 patients in the SIGIR cohort. In the revision, we have added the clarification to the results section. The reasons for choosing the SIGIR cohort for this purpose are two-fold:

(1) We have released the GPT-4 annotations and human judgments on these patient-criterion pairs at <https://huggingface.co/datasets/ncbi/TrialGPT-Criterion-Annotations>.

Keeping the annotations only to the single SIGIR cohort makes sure that future researchers can use the two TREC cohorts as independent test sets for robust evaluations.

(2) We found that the SIGIR cohorts are also annotated with clinician-generated keywords. Therefore, we have compared TrialGPT-Retrieval to them for the first-stage retrieval performance. As such, it keeps the results comparable by annotating the same cohort and comparing the TrialGPT-Matching to clinician eligibility labels for the same patients.

Major Comment 4

Where are the candidate clinical trials in the ranking analysis drawn from? My assumption is that it is just the ~300 trials that were already sampled. I have a significant problem with IR metrics like NDCG and P@10 being used when non-IR methods are used to select the candidates. A major issue with this paper is that the many systems that have been published on these three datasets are not comparable. When one compounds this with metrics that give the appearance of comparability that creates a major point of confusion

(e.g., someone might compare the numbers in Table 2 to some paper published at SIGIR and conclude TrialGPT is better, when in fact the numbers cannot be compared). LLMs generate a lot of hype, and it is scientifically appropriate to ensure information is present that makes it clear what can and cannot be legitimately compared.

Response

Thank you for your thoughtful suggestion. As described in our responses to the Major Comments 1 and 2, we have added a new IR component, TrialGPT-Retrieval, to the TrialGPT framework. In the revision, the candidate clinical trials for the ranking analysis are drawn from TrialGPT-Retrieval instead of directly sampling from the relevance judgments. Due to the addition of the IR component, we keep the using of NDCG and P@10 in the evaluation of TrialGPT-Ranking.

In the revision, we have also added the clarification that the results in Table 2 “*are not directly comparable to the results of TREC CT participating systems as different evaluation settings are used*” to avoid any possible misinterpretation of the results. Specifically, the results are not directly comparable due to the following reasons:

(1) All participating systems only run on a single corpus, while we run on three corpora and report the average performance. As most of these systems are not publicly available, we cannot run their systems on other corpora for comparison.

(2) Most of the top-performing systems use either previous relevance judgments or even in-house annotations to train the systems, while we mainly focus on zero-shot evaluations.

(3) As mentioned in our responses to your Major Comment 2, we used a more realistic initial document collection for the first-stage retrieval, which differs from the ones considered in the competitions. Additionally, the pooled evaluation format used by the SIGIR and TREC datasets inherently biases outcomes in favor of the participating systems.

Major Comment 5

This is not "a systematic evaluation of LLMs", as the Discussion says. It is a comparison of a single LLM to many BERT models. It evaluates TrialGPT across several important dimensions of performance. But it does not systematically evaluate LLMs.

Response

Thank you for your constructive feedback. In the revision, we have removed the claims of “*systematic evaluation*” in the discussion.

Major Comment 6

The authors claim "Our work supports the position that AI models for clinical trial matching should not be designed to replace human recruiters but to empower them...". Yet, nothing in this manuscript supports this claim. The authors are correct to point out

that fully-automated trial matching is not appropriate, but I don't think anyone reading this paper is going to assume that if TrialGPT flags a patient then they will automatically be enrolled. So if the authors believe that one of their experiments actually supports this claim, then by all means justify the statement. Otherwise, it would be best to formulate this opinion as a counterfactual (e.g., "Our work does not justify the position that clinical trial matching should be fully automatic and exclude human recruiters...", but phrase is however you wish).

Response

Thank you for your constructive comment. In the revision, we have removed the original claim and included the suggested opinion: *"Our work does not justify the position that clinical trial matching should be fully automatic and exclude human recruiters. Experts should always be in the loop of medical AI deployments, and the TrialGPT matching results are only used to assist them for improved efficiency."*

Major Comment 7

Finally, I feel that the related work here is not particularly well-cited. There's been a lot of work in this space recently. I realize this manuscript started as a July 2023 preprint, but there has been a ton of work since then and I only noticed four additional citations related to this area (two of which are self-cites). So the authors need to do better at identifying

relevant work and describing the particular gap that this paper fills. Here's an incomplete list:

<https://arxiv.org/abs/2402.05125>

<https://pubmed.ncbi.nlm.nih.gov/37952206/>

<https://arxiv.org/abs/2401.01566> (not to mention the other TREC participants, for which LLMs were highly utilized)

<https://arxiv.org/abs/2312.09958>

This is only a cursory search, of course. This area has consistently been mentioned as one of the biomedical areas that LLMs can make the most immediate impact, so papers are coming out thick and fast and it is important for a journal like this to make the case for why this approach is impactful.

Response

Thank you for mentioning these relevant studies. As you correctly pointed out, our initial manuscript was finished in July 2023, and these papers were not public then. In fact, two of the four studies mentioned in the comment (i.e., <https://arxiv.org/abs/2402.05125> and <https://arxiv.org/abs/2312.09958>) have both cited this work. The former study mentioned “*TrialGPT was the first end-to-end LLM-based system for processing both unstructured clinical text and eligibility criteria (Jin et al., 2023; this manuscript), and achieved high accuracy*”, and latter study is a direct adaptation of our TrialGPT framework: “*Building upon the work presented in [18](Jin et al., 2023; this manuscript), our research extends this*

framework to incorporate and evaluate more opensource LLMs". The fact that **our framework has already been adopted by others shows the impact of this work.**

In the revision, we have added the AutoCriteria study to the "*enhance the first-stage retrieval of clinical trials through information extraction*" sentence. <https://arxiv.org/pdf/2402.05125> (Wornow *et al.*) tackles the trial-to-patient matching task (so it uses the n2c2 2018 cohort instead of the TREC cohorts for evaluation), while our work targets patient-to-trial matching. We have added this to the patient-to-trial matching sentence in the Introduction. In the revision, we have also added the citations to "*Team IELAB at TREC Clinical Trial Track 2023: Enhancing Clinical Trial Retrieval with Neural Rankers and Large Language Models*" and "*Distilling Large Language Models for Matching Patients to Clinical Trials*" as demonstrations of data augmentation with synthetic data generated by large language models.

Minor Comment 1 (Abstract)

-- "first-of-its-kind" is a difficult to assess claim, especially since we don't have good ways of defining the official definition of a "large" language model. But, regardless, some of the TREC CT 2023 participants used LLMs and that was back in August of 2023, so probably the best thing to do is to remove this term.

-- "184 patients" -> "184 synthetic patients"

Response

Thank you for your suggestions on the wording. In the revision, we have replaced “*first-of-its-kind*” with “*novel*”. We have also added “*synthetic*” to highlight that these are not actual patient notes.

Minor Comment 2 (Introduction)

-- Citation #6 looks wrong.

Response

In the revision, we have fixed citation #6 (now citation #8) with the correct conference information: Proceedings of the Twenty-First Text REtrieval Conference (TREC 2012). We have also unified all citations to TREC papers in this format.

Minor Comment 3 (Introduction)

-- "184 patients" -> "184 synthetic patients"

Response

In the revision, we have also added “*synthetic*” to highlight that these are not actual patient notes.

Minor Comment 4 (Introduction)

-- Figure 1c - it isn't clear what the little pictures are indicative of... it gives the sense that TrialGPT is explicitly considering the patient's genes, pill meds, injected meds, and microbiology cultures. Of course TrialGPT doesn't do this, but point is that it is strange to understand.

Response

The icons (genes, pill meds, injected meds, etc.) in the original Figure 1c were intended to represent different clinical trials, corresponding to different ranks and excluding status. In the revision, we have incorporated a brand-new TrialGPT-Retrieval component and redrawn Figure 1. In the new version of Figure 1, clinical trials are clearly denoted by texts, and no icons are used to minimize potential confusion.

Minor Comment 5 (Results)

-- Only two citations for three datasets (appears that TREC CT 2022 isn't cited). Properly citing scientific data is a major focus at the moment.

Response

Thank you for your comment. In the revision, we have added the citation to the Overview paper of TREC Clinical Trials (CT) 2022. In addition, we have also added the citations to TREC Clinical Decision Support (CDS) Tracks 2014 and 2015, based on which the SIGIR cohort was built, to acknowledge the topic sources.

Minor Comment 6 (Results)

-- "Interestingly, patients meet more exclusion..." -> "Importantly, patients meet more exclusion" (it isn't interesting, it is a well-known and obvious fact to anyone that understands clinical trials, but it is important and worth the authors pointing out)

Response

In the revision, we have replaced “*Interestingly*” with “*Importantly*” as suggested.

Minor Comment 7 (Discussion)

-- "We also notice... might be over-simplified... (geolocations, recruitment status, etc.)" --
It is of course appropriate to point out the aspects of clinical trial search that are not covered by the datasets utilized, but this wording suggests the authors have noticed something that isn't fully explicit from the dataset creators. The dataset creators do in fact make this explicit and (in their minds) exclude geolocation/recruitment status with good reason. Also, "etc." implies there are likely other things, but the dataset creators only explicitly excluded geolocation and recruitment status from consideration, so if the authors have noticed additional issues not mentioned by the dataset creators, please be explicit about this. Otherwise it would be more appropriate to phrase this sentence along the following lines: "Further, the SIGIR and TREC datasets focus on the semantics of the clinical trial inclusion/exclusion criteria, excluding factors such as geolocation and trial recruitment status as these are addressable through traditional structured query

approaches. As a result, any use of TrialGPT would similarly need to ensure the identified trials are appropriate for a patient along these lines as well."

Response

Thank you for your thoughtful comment. In the revision, we have replaced the original sentence with the suggested sentence: *"Further, the SIGIR and TREC datasets focus on the semantics of the clinical trial inclusion/exclusion criteria, excluding factors such as geolocation and trial recruitment status as these are addressable through traditional structured query approaches. As a result, any use of TrialGPT would similarly need to ensure the identified trials are appropriate for a patient along these lines as well."*

Minor Comment 8 (Discussion)

-- "GPT-4 is the most capable" -> "GPT-4 is currently the most capable"

Response

In the revision, we have added the suggested *"currently"* to be more precise.

Minor Comment 9 (Discussion)

-- "our pilot user study is of limited scope in sample size" -> state the sample size, e.g., "(n=XX)".

Response

In the revision, we have added the sample size accordingly “(N=183)”.

Comment

I briefly reviewed the data files, which to some extent are more important in my mind since some of the issues I had when reading the manuscript (about data selection) wasn't well explained. But at least the data uploaded allows for some understanding.

I'm not 100% happy that the data simply takes the SIGIR/TREC data directly, since oftentimes this results in this paper being cited in lieu of the original dataset. But seeing as how I am the creator of 2 of the 3 datasets (and actually created the topics used in the SIGIR dataset, if not the trial annotations), it feels odd to point this out.

Response

Thank you for your comments on our manuscript and your important resource contributions to open science! As a community, we all benefit from these publicly available datasets. In the revision, we have added more descriptions of how they are re-used for the evaluation of TrialGPT, please see our responses to your Major Comments 1, 2, and 3. We have also added the citations to the TREC CDS 2014 and 2015 cohorts in the description of the SIGIR cohort to acknowledge the topic sources. In addition, we have included an explicit instruction in our released code repository (<https://github.com/ncbi-nlp/TrialGPT>) that requests proper citations to the original dataset papers:

If you use the SIGIR cohort, please cite the original dataset papers by:

```
@inproceedings{koopman2016test,  
  title={A test collection for matching patients to clinical trials},  
  author={Koopman, Bevan and Zuccon, Guido},  
  booktitle={Proceedings of the 39th International ACM SIGIR conference on Research and Develo  
  pages={669--672},  
  year={2016}  
}  
@inproceedings{roberts2015overview,  
  title={Overview of the TREC 2015 Clinical Decision Support Track},  
  author={Roberts, Kirk and Simpson, Matthew S and Voorhees, Ellen M and Hersh, William R},  
  booktitle={Proceedings of the Twenty-Fourth Text REtrieval Conference (TREC 2015)},  
  year={2015}  
}  
@inproceedings{simpson2014overview,  
  title={Overview of the TREC 2014 Clinical Decision Support Track},  
  author={Simpson, Matthew S and Voorhees, Ellen M and Hersh, William R},  
  booktitle={Proceedings of the Twenty-Third Text REtrieval Conference (TREC 2014)},  
  year={2014}  
}
```

If you use the TREC cohorts, please cite the original dataset papers by:

```
@inproceedings{roberts2021overview,  
  title={Overview of the TREC 2021 clinical trials track},  
  author={Roberts, Kirk and Demner-Fushman, Dina and Voorhees, Ellen M and Bedrick, Steven and  
  booktitle={Proceedings of the Thirtieth Text REtrieval Conference (TREC 2021)},  
  year={2021}  
}  
@inproceedings{roberts2022overview,  
  title={Overview of the TREC 2022 clinical trials track},  
  author={Roberts, Kirk and Demner-Fushman, Dina and Voorhees, Ellen M and Bedrick, Steven and  
  booktitle={Proceedings of the Thirty-first Text REtrieval Conference (TREC 2022)},  
  year={2022}  
}
```

Reviewers' Comments:

Reviewer #1:

Remarks to the Author:

The revision has addressed all my concerns. I particularly like how the authors introduced a LLM-based retrieval step and evaluated e2e comparison against manual trial matching. One remaining suggestion: TREC datasets are useful benchmarks, but they tend to be small given the cost in annotation. The paper can benefit from elaborating discussion on potential growth areas in evaluation and how likely the proposed method can fare in real-world scenarios.

Reviewer #2:

Remarks to the Author:

I appreciate the authors for addressing my comments, and I value the addition of TrialGPT-Retrieval to filter out irrelevant trials, making the proposed method more scalable and practical. The latest version of the manuscript is clearly written and comprehensive. I have no further comments.

DEPARTMENT OF HEALTH & HUMAN SERVICES

Public Health Service

National Institutes of Health
National Library of Medicine
Bethesda, Maryland 20894

Aug 21, 2024

Dear Reviewers,

Thank you for your positive feedback on our revised manuscript. We sincerely appreciate the time and effort you invested in reviewing our work. Your comments have been vital in enhancing the quality and clarity of our manuscript. In the following sections, we provide a point-by-point response to each of the comments raised by the reviewers, alongside the corresponding changes made in the manuscript.

Response to Reviewer #1

Comment

The revision has addressed all my concerns. I particularly like how the authors introduced a LLM-based retrieval step and evaluated e2e comparison against manual trial matching. One remaining suggestion: TREC datasets are useful benchmarks, but they tend to be small given the cost in annotation. The paper can benefit from elaborating discussion on potential growth areas in evaluation and how likely the proposed method can fare in real-world scenarios.

Response

Thank you very much for your comments. In the discussion section, we have described the real-life considerations of the patient-to-trial matching task that is different from the TREC datasets: “This requires the model to (1) attend to much longer contexts, (2) process structured data, and (3) process multi-modal inputs. These aspects have not been evaluated by this study but are worth exploring in future work”. In the revision, we have further elaborated this point in the same paragraph: “Additionally, future evaluations should investigate how TrialGPT performs when integrating data from electronic health records (EHRs), which often include a combination of structured and unstructured data sources. The ability to seamlessly incorporate such diverse data types would significantly enhance the real-world applicability and increase the validation sample size of our framework”.

Responses to Reviewer #2

Comment

I appreciate the authors for addressing my comments, and I value the addition of TrialGPT-Retrieval to filter out irrelevant trials, making the proposed method more scalable and practical. The latest version of the manuscript is clearly written and comprehensive. I have no further comments.

Response

Thank you very much for your comments.